# Ocean cross-validated observations from the R/Vs L'Atalante, Maria S. Merian and Meteor and related platforms as part of the EUREC[4]A-OA/ATOMIC campaign

Pierre L'Hégaret[1], Florian Schütte[2], Sabrina Speich[1], Gilles Reverdin[3], Dariusz B. Baranowski[4], Rena Czeschel[5], Tim Fischer[5], Gregory R. Foltz[6], Karen J. Heywood[7], Gerd Krahmann[5], Rémi Laxenaire[1], Caroline Le Bihan[8], Philippe Le Bot[8], Stéphane Leizour[8], Callum Rollo[9], Michael Schlundt[5], Elizabeth Siddle[7], Corentin Subirade[1], Dongxiao Zhang[10,11], and Johannes Karstensen[5]

[1]LMD/IPSL, CNRS, ENS, École Polytechnique, Institut Polytechnique de Paris, PSL, Research University, Sorbonne Université, Paris, France
[2]Max-Planck-Institute for Meteorology, KlimaCampus, Hamburg, Germany
[3]Sorbonne Université, CNRS, IRD, MNHN, UMR7159 LOCEAN/IPSL, Paris, France
[4]Institute of Geophysics, Polish Academy of Sciences, Warsaw, Poland
[5]GEOMAR Helmholtz Centre for Ocean Research Kiel, Kiel, Germany
[6]NOAA, Atlantic Oceanographic and Meteorological Laboratory, Miami, FL, USA
[7]Centre for Ocean and Atmospheric Sciences, School of Environmental Sciences, University of East Anglia, Norwich, UK
[8]French Research Institute for Exploitation of the Sea (IFREMER), UBO, CNRS, IRD, Laboratoire d'Océanographie Physique et Spatiale (LOPS), IUEM, Plouzané, France
[9]Voice of the Ocean Foundation, Gothenburg, Sweden
[10]Cooperative Institute for Climate, Ocean, and Ecosystem Studies, University of Washington, Seattle, WA, USA
[11]NOAA PMEL, Seattle, WA, USA

**Correspondence:** Pierre L'Hégaret (pierre.lhegaret@univ-brest.fr)

**Abstract.**

The northwestern Tropical Atlantic Ocean is a turbulent region, filled with mesoscale eddies and regional currents. In this intense dynamical context, several water masses with thermohaline characteristics of different origins are advected, mixed, and stirred, at the surface and at depth. The EUREC[4]A-OA/ATOMIC experiment that took place in January and February 2020 was dedicated to assess the processes at play in this region, especially the interaction between the ocean and the atmosphere. For that, four oceanographic vessels and different autonomous platforms measured properties near the air-sea interface and acquired thousands of upper-ocean (up to 400-2000 m depth) profiles. However, each device had its own observing capability, varying from deep measurements acquired during vessel stations to shipboard underway near-surface observations and measurements from autonomous and uncrewed systems (such as Saildrones). These observations were undertaken with a specific sampling strategy guided by near-real time satellite maps and adapted every half day based on the process that was investigated. These processes were characterized by different spatio-temporal scales: from mesoscale eddies, with diameters exceeding 100 km, to submesoscale filaments of 1 km width. This article describes the data sets gathered from the different devices and how the data were calibrated and validated, in order to ensure an overall consistency, the platforms' datasets are cross-validated using a hierarchy of instruments defined by their own specificity and calibration procedures. This has enabled the quantification

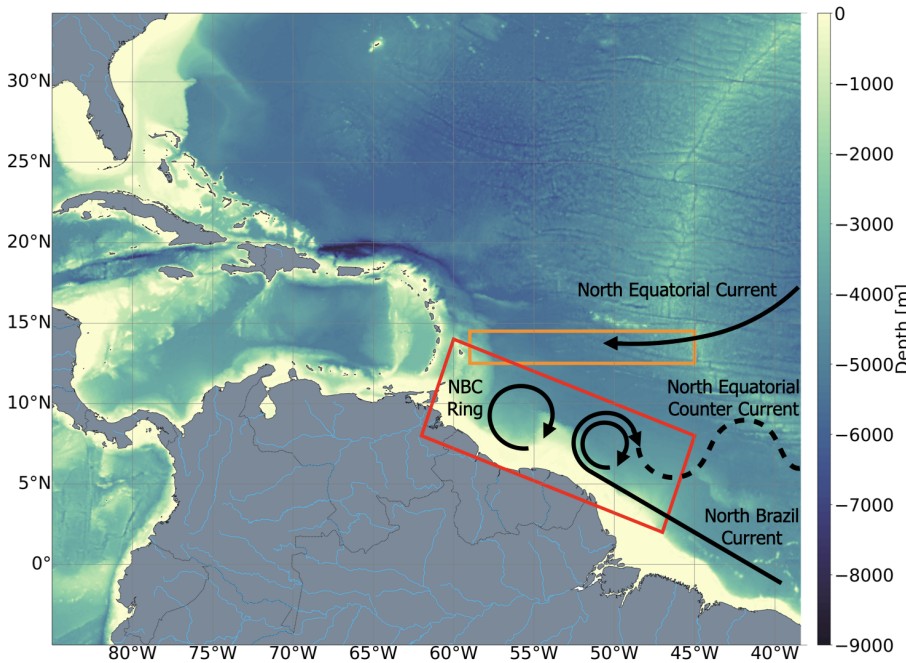

**Figure 1.** Bathymetric map of the northern tropical Atlantic Ocean, with the EUREC$^4$A-OA/ATOMIC regions of interest framed in orange for the "Trade wind alley" and in red for the "Eddy boulevard". The main surface currents of these two regions are schematically represented in black. NBC stands for North Brazil Current.

15  of the uncertainty of the measured parameters when different datasets are used together, e.g. https://doi.org/10.17882/92071 (L'Hégaret et al., 2020a).

## 1   Introduction

The international EUREC$^4$A-ATOMIC initiative (https://eurec4a-oa.eu/) aimed to better understand the link between at-
20  mospheric shallow convection, cloud formation and the general circulation of the atmosphere (Stevens et al., 2021). The EUREC$^4$A-OA experiment was embedded in EUREC$^4$A-ATOMIC, and took place in January and February 2020 in the north-western Tropical Atlantic Ocean. EUREC$^4$A-OA focused on the impact of meso- and submesocale regional ocean dynamics on processes at the air-sea interface. The targeted ocean sampling for EUREC$^4$A-OA was done with four research vessels: the German Maria S. Merian (Karstensen et al., 2020) and Meteor (Mohr et al., 2020), the French L'Atalante (Speich et al.,
25  2021b), and the US Ronald H. Brown (Quinn et al., 2021). In addition, various autonomous platforms, underwater electric gliders, surface drifters, Argo profiling floats, Saildrones and prototype drifting buoys OCARINA and PICCOLO (Bourras et al., 2014) were operated in coordination with the ships.

The EUREC[4]A-ATOMIC experiment took place in a rich dynamical context, where several water masses of diverse origins are advected (see Figure 1, stirred, and mixed, preserving, however, large horizontal and vertical contrasts. Figure 1 from Fratantoni and Glickson (2002) summarizes the main upper-ocean features. The research vessels and platforms deployed during the experiment focused on two subregions: one east of Barbados characterized by a rather stable wind-regime of easterlies ("Trade wind alley"), the other to the south and bounded by the South American continent, a region that hosts intense and long-lived northwestward-drifting mesoscale eddies spawned by North Brazil Current (NBC) retroflection ("Eddy boulevard"). Also, along the shelf break, the Amazon River plume flows northward and actively interacts with the North Brazil Current and its mesoscale eddies, the NBC Rings Fratantoni and Glickson (2002); Fratantoni and Richardson (2006). The evolution, characteristics, and inter to intra-annual variability, of NBC Rings are important elements of the global ocean circulation. As the North Brazil Current sheds eddies that move northward along the South American continental slope, they provide an essential part of the interhemispheric transport of mass, heat, salt, and many different biogeochemical ocean properties, thus having a key role in the Atlantic Meridional Overturning Circulation (AMOC; Johns et al. (2003)). In this study, we will use the devices deployed by the R/V Meteor, that mainly focused on the Trade wind alley, and by the R/Vs L'Atalante and Maria S. Merian, that sampled the Eddy boulevard. We will focus on the oceanic measurements (temperature, salinity, oxygen, and velocity) and leave aside the atmospheric and meteorological ones, some of which are already discussed in companion papers in this Journal special issue dedicated to EUREC[4]A Bailey et al. (2023); Bosser et al. (2021); Stephan et al. (2021) and others will be submitted soon.

For observational studies of meso-/submesoscale features, some means of adaptive sampling is required. The EUREC[4]A-OA sampling was guided by analysis of near-real time satellite maps of Sea Surface Temperature, Sea Surface Salinity, altimetry (absolute dynamic height and related geostrophic velocities) and ocean color (Speich et al., 2021b). Guided by the satellite information, the surveys were then done by various observational platforms in order to sample specific features, such as mesoscale eddies and fronts, freshwater pools, and filaments. To address the anticipated spatio-temporal sampling, the various platforms with different resolutions, autonomy, sensor payloads (Liblik et al., 2016), and periods required to acquire a profile, were used in a concerted effort (Figure 2). The sampling strategy was designed such that the phenomena would be measured with sufficient temporal and spatial resolution while also paying attention to the synchronicity of observations in the ocean and atmosphere.

In order to ensure interoperable data for a parameter measured with various sensors, a comparative quality analysis was also performed, known as secondary quality control (QC). Secondary QC aims to create a coherent data set out of various data streams.

Gouretski and Jancke (2000) were among the first to present secondary quality control on oceanographic data through the use of "crossover analysis" in deep waters. This enabled them to conduct a rigorous QC assessment by comparing data from different sources and determine systematic errors such as Standard Seawater batch offsets. As for other similar studies (e.g.,

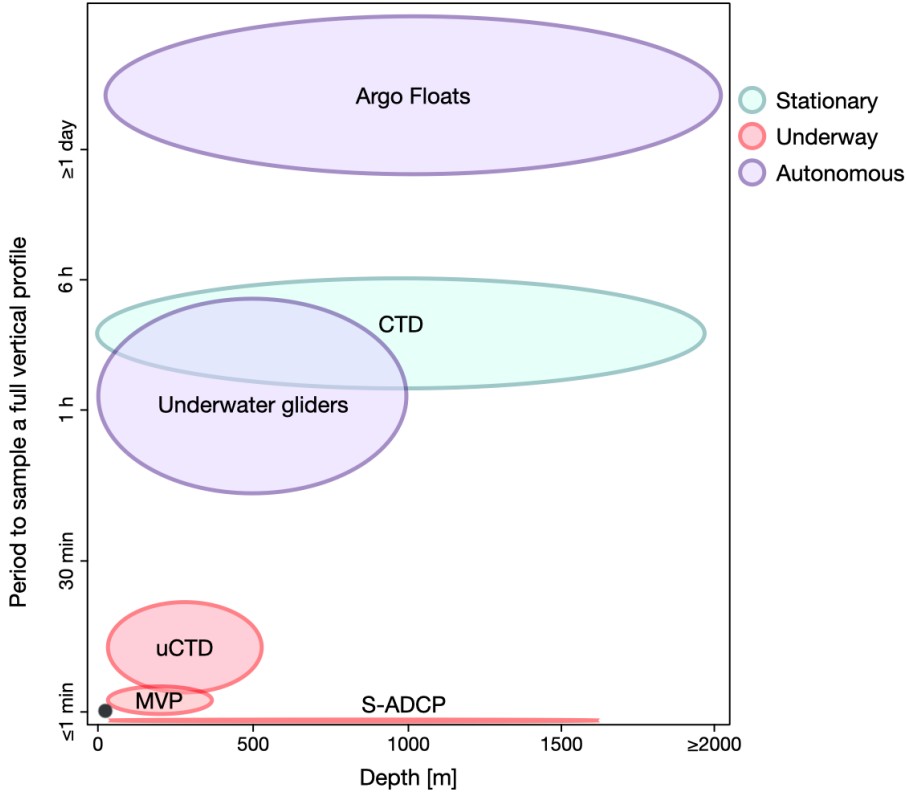

**Figure 2.** Schematic representation of the devices deployed during the EUREC[4]A-OA experiment, characterized by their frequency of acquisition for a full vertical profile and depth reached, considering the time of deployment. The green color indicates observations undertaken with the ship on station; the red color designates the devices acquiring data underway, whereas the purple color represents autonomous platforms. The dot in the lower left corner represents the surface measurements made by the ships' TSG, Saildrones and surface drifters.

Tanhua et al. (2010)), a basic assumption is that deep water (typically >2000 m) properties are close to invariant. The need for QC methods in the upper layer received much attention through the global operations of profiling floats that typically are not recovered, and hence receive direct sensor QC only before deployment (e.g., in lab and in reference to standard or reference material). As introduced by, for example, Wong et al. (2003), by comparing with all nearby profile data and considering distance in space and time as a measure for impact, the offset, and drift behaviors of specific float measurements can be reconstructed.

The goal is to perform a quality assessment of a dataset, which in turn is composed of individual datasets from different observations platforms and sensors. For this purpose, we introduce here an assessment scheme of the individual datasets based on their traceability to reference data (traceability level). In the most optimal case, the reference material (RM) is a defacto standard (Otosaka et al., 2020), such as standard seawater for salinity, oxygen titration for dissolved oxygen, or triple point

cells for temperature. As an example, for salinity the RM is Standard Seawater (SSW), giving a hierarchy as follows: SSW is assigned traceability level 0 and is used to reference a salinometer (Bacon et al., 2007) (traceability level 1). The salinometer readings for bottle samples are used as a statistical basis for the ship's CTD salinity correction (traceability level 2). The corrected ship CTD salinity is then further used to correct the TSG (traceability level 3). Bottle samples were also collected for the TSG and used via the salinometer as a statistical basis for the ship's CTD salinity correction (traceability level 2). The TSG is used to correct the Saildrones data (traceability level 4 or level 3, depending on the TSG calibration). Similar hierarchies can be built for all sensors.

Part of the Quality Assurance of the combined dataset is determining accuracy and precision, and also considering in this process the expected stability of the sensors given by the manufacturer. The CTD Rosette is key in the calibration process in respect of the secondary quality control. From the water samples collected with the Rosette sampler, numerous variables (in addition to conductivity, temperature, and pressure) are accessible throughout the water column, used for sensor calibration, laboratory analysis (e.g., oxygen titration), and to analyze biogeochemical parameters. However, its deployment requires the ship to remain on station for a few hours (depending on the attained depth) to perform the vertical profiles. The underway CTD (uCTD) and Moving Vessel Profiler (MVP) casts can be carried out underway, but they cannot dive deeper than about 450 m for the uCTD and about 200 m for the MVP model we employed (MVP30-300) (Branellec et al., 2020; Karstensen et al., 2020; Speich et al., 2021b). Also, their sensors are more subject to bias than the CTD (Ullman and Hebert, 2014). Other devices, such as underwater gliders, drifters, and Argo floats, are autonomous, but the data calibration is limited to certain times (deployment and recovery as shown on Figure 2) or by comparing sensor readings from nearby (time and space) data that has passed a QC. All the underway and autonomous devices rely on the proximity in space and time of CTD casts to validate and calibrate their measurements.

In the following section, we present the calibration strategy adopted in this study, as well as the hierarchy of traceability between the sensors of each device. In section 3, we present the ship observations, providing information on when and where they were deployed, and how they were calibrated and validated. There, we focus on the CTD and how they are used to estimate the uncertainty of platforms with lower level of traceability. Section 4 is constructed similarly, but focusing on autonomous devices. In section 5 we provide an example of data concatenation of different platforms. Finally, we describe the final data set and the variables we provide for use to the scientific community.

## 2 Calibration strategy

The EUREC[4]A-OA experiment relied on numerous devices to measure physical and chemical properties of the water column from the three European ships and the various autonomous platforms deployed from the ships. We also include here the five US Saildrones funded under the NOAA and NASA ATOMIC project (Quinn et al., 2021; Gentemann et al., 2020). Their deploy-

ments were conducted by scientific staff originating from two institutions: GEOMAR (Kiel, Germany) and IFREMER (Brest, France). They have specific practices, sometimes leading to different procedures of deployment, data acquisition and calibration, while still complying to international standards (Sloyan et al., 2019). The various calibration practices are either linked to similar devices from different manufacturers, or to various procedures in laboratories before and after the cruise. All the devices deployed during EUREC[4]A-OA are commonly used in oceanographic cruises, and the sources of errors and calibration procedures have been extensively documented and studied. In the next section, we briefly summarize them for each device.

At the top level of our hierarchy stands the most traceable sensors used to read water samples issued from the CTD bottles during every vertical profile or from the TSG circuit.

The CTD measurements are at the second level of the hierarchy, as its sensors went through careful pre-cruise and post-cruise calibration at the manufacturer's facility. Moreover, the sensor measurements are carefully validated and calibrated with samples collected and analyzed from the Rosette bottle water samples. CTD measurements serve as references for the calibration of all other observing platforms. Some TSG underway measurements also stand at traceability level 2 when calibrated with bottle samples.

Next in the ranking (traceability level 3) come those devices whose measurements are calibrated with the CTD, as they sample the same water (devices located on the same ship from which the CTD was deployed and measuring the same water as the CTD). These are the TSG when not directly calibrated with bottle samples, and the uCTD sensors when the probes were purposely mounted for calibration on the CTD rosette. The main sources of error here come from the method of deployment of the devices and the sampling rate of the sensors.

Sensor data that only are calibrated via reference data close in space and time to the CTD profiles are labelled traceability level 4. The cross-calibration is then achieved by comparing the vertical profiles on T/S and depth/density diagrams. In this category fall most of the underway profiling devices: uCTD, MVP, and underwater gliders. As these measurements are not synchronous and not co-located with CTD profiles, an additional source of uncertainty arises from the spatio-temporal ocean variability.

At the bottom of the hierarchy of traceability to a standard or an RM are the devices that cannot be compared directly either because their sensors are specific, measuring quantities that are not directly comparable with the CTD or that can be calibrated only by the manufacturers.

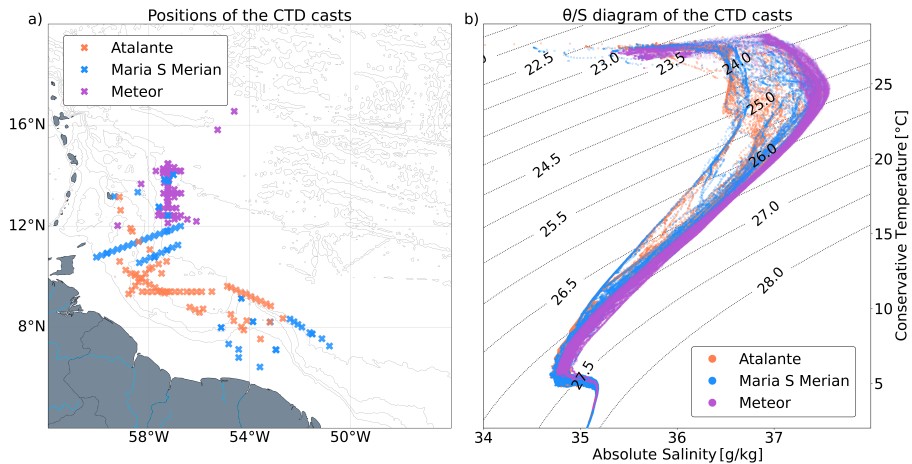

**Figure 3.** a) Map of the CTD casts positions for the R/Vs L'Atalante, Maria S. Merian and Meteor. b) θ/S diagram of the CTD profiles for each ship superimposed on the isopycnals.

## 3 Ship observations

### 3.1 CTD rosette

On the R/Vs Maria S. Merian, Meteor, and L'Atalante a SeaBird SBE911+ CTD system was used for high-quality vertical profiling of the water column. Hereafter, we will describe briefly the calibrations and validations for each sensor. Other sensors were mounted to the CTD device (e.g., fluorescence, turbidity, particles) but are not considered here. Similar operational practices were carried out on all ships for CTD profiling. First, the CTD system was lowered to shallow depth ( 5 m) until the pump started. Then the CTD was brought back to the surface and subsequently lowered at approximately 0.5 m/s in the first 100 meters and 1 m/s for the deeper water column. The target depth varied, but in most cases reaching just above the seafloor. Water samples were collected with Niskin bottles mounted to the CTD rosettes. The samples were used for sensor calibration and for further analysis. The procedure for closing the bottles differed between the ships: on the R/V L'Atalante the CTD was stopped a few seconds for sampling, while on the R/V Maria S. Merian the samples were taken without stopping the CTD package.

In total the number of profiles acquired were 64, 86, and 266 for the R/Vs L'Atalante, Maria S. Merian and Meteor respectively (Figure 3).

The CTD rosettes on R/Vs Maria S. Merian and L'Atalante were also equipped with two Lowered Acoustic Doppler Current Profilers (L-ADCP), in a system composed of one instrument looking upward and the other one looking downward, to record ocean current profiles.

### 3.1.1 Pressure, Temperature, Conductivity, and Salinity quality assurance

160 Based on the manufacturer (SeaBird) specifications, the SBE9+ probe measures pressure with an initial accuracy of $\pm 1.5$ $10^{-2}\%$ and a resolution of $\pm 1\ 10^{-3}\%$ of the full scale of the respective CTD (6800 m for the R/Vs Maria S. Merian and Meteor).

For the R/V L'Atalante, the pressure sensors are calibrated before and after the cruise at the IFREMER Laboratory of 165 Metrology. The calibration is performed at a constant temperature of 20 °C with increasing and decreasing pressure levels, with an uncertainty of 0.6dbar at 2000 dbar. The bias, measured during the calibration, is then corrected by a polynomial of degree 4, associated with an uncertainty of $\pm 0.12$dbar. Good stability of the sensor was observed, with an overall uncertainty of $\pm 0.72$dbar. In addition, a validation was made using reversing pressure meter sensors (SIS RPM 6000X) at the bottom of the profiles and by comparing them with the CTD sensor to assess any drift. No drift were observed during the cruise, so after 170 laboratory calibration and in-cruise validation, we assign the CTD pressure sensor level 2 traceability and an uncertainty on the order of the initial sensor accuracy of $\pm 1.5\ 10^{-2}\%$. For the pressure sensors of the R/Vs Maria S. Merian and Meteor, no dedicated lab calibration are carried out.

For all ships, the pressure sensor offset on deck before and after each profile was corrected in the processing as an offset, 175 typically the mean of all values for each probe.

For each CTD, two SBE3+ temperature sensors from Seabird were mounted on the SBE911 probe, and the most stable sensor was used for the final calibration. The accuracy and resolution provided by the manufacturer are of $\pm 1\ 10^{-3}$ °C and $\pm 2$ $10^{-4}$ °C.

180

For the R/V L'Atalante, the temperature readings were calibrated at the IFREMER metrology lab before and after the cruise in reference to a Rosemount-type platinum resistance, periodically checked and certified, in a bath with strictly controlled temperature. The measurements were corrected by applying a polynomial of degree 3. The maximal error is lower than the sensor accuracy provided by the manufacturer. In addition, the temperature sensor stability was monitored in comparison with 185 two reversing thermometers (SIS RTM 4002X), one closed at the deepest depth of the profile, the other during the descent. No drift was observed during the cruise. For the R/Vs Maria S. Merian and Meteor, the sensors were calibrated before the cruise at an authorized lab.

For the temperature sensors from the CTD we thus assume an uncertainty of $1\ 10^{-3}$ °C, corresponding to the manufacturer 190 accuracy, and we assign them level 2 traceability in our hierarchy.

Conductivity is measured with two Seabird SBE4 sensors and, as for temperature, the most stable sensor is used for final calibration. In general, the procedures followed the GO-SHIP recommendations in Hood et al. (2010), but details are provided below. The accuracy of the SBE4, as provided by the manufacturer, is $\pm 3\ 10^{-3}$ mS/cm with a nominal stability of $3\ 10^{-4}$ per month and a resolution of $\pm 4\ 10^{-5}$ mS/cm at 24Hz sampling.

For sensors used on the R/Vs, a lab calibration was done before the cruise by a manufacturer (Seabird) authorized laboratory. During the cruises, water samples were collected from Niskin bottles to perform a CTD conductivity calibration. Salinity of the water samples was analyzed on the R/V Maria S. Merian using an Optimare salinometer and on the R/V L'Atalante using a Portasal salinometer. The salinometers were in turn calibrated against a reference material, Standard Seawater. On the R/V Maria S. Merian a secondary reference also was used (labelled "substandard"), which is a large volume of water with unknown but constant salinity. At regular intervals, the substandard was measured with the salinometer and tracked for stability as an indicator of potential drift of the salinometer without the need to use large amounts of Standard Seawater. One other slight difference in procedures was the treatment of the water samples before analysis with the salinometer. In addition to adjusting the samples to the laboratory temperature (R/Vs Maria S. Merian and L'Atalante), the samples were degassed on the R/V Maria S. Merian. On earlier cruises, it was found that the Optimare salinometer is more sensitive to gas bubbles. For the purpose of degassing, the bottles were heated in a water bath to 5-10 °C above laboratory temperature and then brought back to laboratory temperature. The released gas was extracted. Only then were the samples analyzed. More information on the CTDs and salinometers can be found in the cruise reports (Karstensen et al., 2020; Branellec et al., 2020). The salinity data from bottle samples of the R/Vs Maria S. Merian and L'Atalante is considered level 1 traceability.

The processing of CTD conductivity was done by first applying basic processing steps from the SBE processing routines (Seasoft V2) and including loop edit (0.2 m sec). The prepared raw data was then calibrated using the bottle sample analysis. Slightly different approaches were taken for the R/Vs Maria S. Merian and L'Atalante.

The R/V Meteor did not have a salinometer on board. Samples were collected and stored onboard for later analysis on shore. However, the number of samples available was too small to perform a meaningful statistical analysis. At two occasions, CTD profiles from the R/Vs Maria S. Merian and Meteor were performed close in space (1500 and 50 m apart) and covered the full water column. They are used here for the purpose of the R/V Meteor CTD sensors calibration and validation. A -4db offset was found between the profiles. As a consequence, we applied an equivalent correction to the R/V Meteor measurements. This led us to consider the R/V Meteor CTD data at level 3 in traceability.

For the R/V Maria S. Merian, the processing was as follows: after allocating a downcast profile segment to the upcast bottle sample stop via a vertical gradient criterion, the conductivity difference between bottle sample analysis and CTD sensor recording was calculated. The differences were sorted by magnitude, and the first 33% of all values were removed to eliminate outliers. Based on the remaining 66% of all values, correction equations were derived using pressure, conductivity, time, and

sample number. Care was taken not to use high-order equations, as spurious interpolation may appear for weakly constrained segments of the multiparameter fit space. The uncertainty of the R/V Maria S. Merian sensors is estimated to be $2 \times 10^{-3}$ psu, and the data are assigned level 2 traceability. A similar procedure was used for the oxygen sensor calibration based on the results from the Winckler titration.

For the R/V L'Atalante a set of three corrections was applied to remove large differences between the conductivity values of the sensor and water sample. First, a correction as a function of time was implemented to take into account a potential slow drift of the conductivity sensor. Second, a correction relative to the conductivity was applied. At each iteration of this correction, the samples showing $\Delta C > 2.8 \times \sigma$, $\Delta C$ being the difference between the sensor and the water sample conductivity, and $\sigma$ the standard deviation of all the samples considered at each iteration, were removed. Third, a correction as a function of pressure was applied to the conductivity or salinity. After calibration of all casts, the standard deviation between the sensor data and the chemical data was $2.3 \times 10^{-3}$ mS/cm for conductivity and $2.3 \times 10^{-3}$ psu for salinity, both below the accuracies provided by Seabird. The uncertainty of the R/V L'Atalante sensors is thus of $3 \times 10^{-3}$ psu, and they are considered level 2 traceability.

### 3.1.2 Dissolved Oxygen

Two SBE43 dissolved oxygen sensors were used on the R/Vs L'Atalante and Maria S. Merian, for a range of measurements from 0 to 120% of the surface saturation. The accuracy from the manufacturer is 2% of the saturation. The sensor showing the more stable measurements was kept for data reduction.

Pre- and post-cruise lab calibrations were carried out on the sensors in laboratories in the same way as the temperature and conductivity sensors. As for conductivity, water samples were collected in bottles for calibration of the sensor measurements. The dissolved oxygen concentrations in the water samples were estimated using Winkler titration (Winkler, 1888). The chemistry reports for the different R/Vs describe the operating modes for the R/V L'Atalante (Branellec et al., 2020) and R/V Maria S. Merian (Karstensen et al., 2020).

After calibrations, the uncertainties of the oxygen measurements are 1.60 $\mu$mol/kg for the R/V L'Atalante and 0.61 $\mu$mol/kg for the R/V Maria S. Merian. No CTD measurements nor samples were collected for the R/V Meteor. The CTD oxygen data are considered level 2 traceability.

### 3.1.3 Lowered ADCP (L-ADCP)

For every CTD station on the R/V L'Atalante and Maria S. Merian, two Workhorse 300 kHz ADCP were attached to the CTD rosette, one looking upward and the other looking downward. They provide current profiles from the surface to the maximum depth of the CTD cast. Reference velocities to derive velocity profiles from the velocity shear observations of the L-ADCP system were obtained from the ship ADCP (S-ADCP) and the bottom track (if available) following Thurnherr et al. (2010) and

Sloyan et al. (2019). The accuracy of L-ADCP velocity measurements is estimated to be $\pm 0.5$ cm/s. The velocity measurements are specific in our hierarchy of calibrations, since they can only be calibrated with cross-validation between devices and not with water samples. Therefore, we rank them level 2 traceability in the hierarchy of sensor/platform quality assurance.

### 3.1.4 Nutrients and Bio-optical measurements

While the previous sensors and procedures are similar for the R/Vs Maria S. Merian and L'Atalante, this is not the case for the measurement of nutrients and other biogeochemical properties. Following the recommendation from GO-SHIP (Sloyan et al., 2019), numerous bio-optical sensors were also mounted on all CTD rosettes. The various sensors had very different and sometimes unknown lab or manufacturer calibrations. They are considered level -9, the lowest in our calibration hierarchy.

The CTD rosette onboard the R/V Maria S. Merian was equipped with an OPUS UV spectral sensor for nitrate and carbon bond measurements, more specifically nitrate (NO3-N), nitrite (NO2-N) and numerous organic ingredients with a resolution of 0.8 nm/pixel using wavelengths of 200-360 nm. In addition, on all three ships, nutrients were analyzed from the bottle samples taken. On the R/V L'Atalante for each CTD cast, three bottles collected samples at different depths to measure Phosphate, Silicate, Nitrate, and Nitrite concentrations after the cruise. For the R/Vs Meteor and Maria S. Merian, these quantities, as well 275 as ammonium, were measured on specific stations and at fixed depths between the surface and 350 meters.

Chlorophyll fluorescence was measured via fluorometers. The CTD deployed from the R/V L'Atalante was equipped with a Chelsea AquaTracka III. The accuracy provided by the manufacturer is $\pm 2 \ 10^{-2}$ $\mu$g/L and its sensitivity is $1 \ 10^{-2}$ $\mu$g/L. The R/Vs Meteor and Maria S. Merian used Wet Labs Eco-AFL/FL fluorometers. Their sensitivity is $2.5 \ 10^{-2}$ $\mu$g/L. The 280 CTD rosettes of the two German ships were also equipped with WET Labs Eco CDOM fluorometers, measuring the dissolved organic matter with a sensitivity of $9.3 \ 10^{-2}$ ppb.

The CTD deployed from the R/V L'Atalante was instrumented with a C-Star transmissometer from Wet Labs, measuring particle beam attenuation coefficient.


The CTDs of the R/Vs Maria S. Merian and Meteor were also equipped with turbidity meters (WET Labs, Eco-NTU) to measure the turbidity of water with a sensitivity of $2 \ 10^{-2}$ NTU (Nephelometric Turbidity Units) in the upper 125 meters of the water column, and 0.12 NTU down to 1000 meters depth.

Finally, all three ships CTDs were equipped with PAR/Irradiance, Biospherical/Licor from Chelsea. This sensor measures the number of photons in the 400–700 nm wavelength, the spectral range of Photosynthetically Active Radiation (PAR), converted in mMol/s/m$^2$. Additionally, surface PAR/Irradiance sensors were mounted on both the R/Vs L'Atalante and Meteor

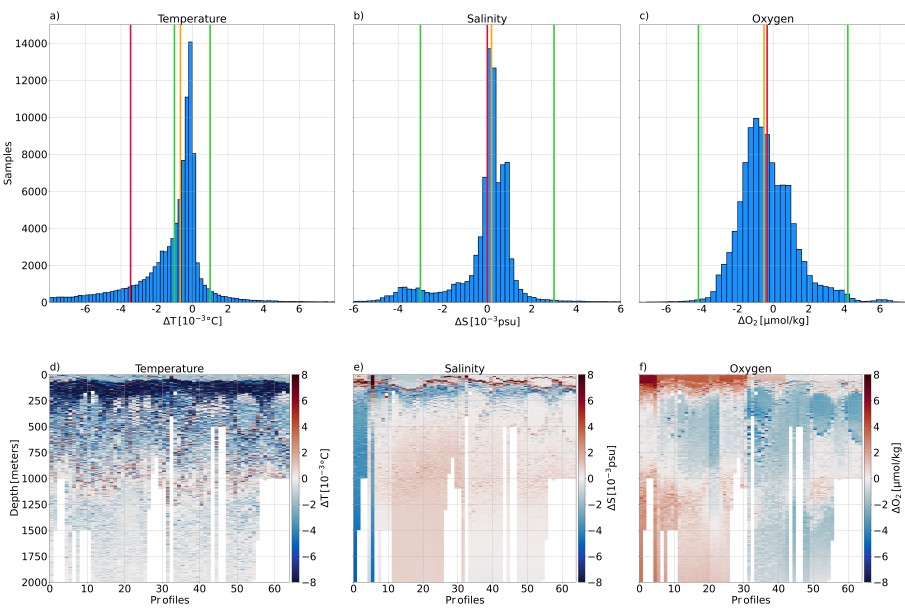

**Figure 4.** a), b) and c) Histograms displaying the differences respectively in temperature, salinity and oxygen between the IFREMER and GEOMAR calibrations applied to the R/V L'Atalante CTD profiles. Green lines indicate the accuracy claimed by the manufacturer, red and orange lines show the mean and median differences. d), e) and f) display these differences vertically for each CTD profile.

CTD rosettes.

As these measurements (fluorescence, transmissometer, turbidity, and PAR) were not validated and calibrated, we did not add them to our hierarchy, nor did we perform a secondary calibration between devices.

## 3.2   Intercalibration of CTD data – the effect of oceanic variability

For all three ships, the acquisition of temperature, salinity, and dissolved oxygen data are performed with sensors that have sim-
ilar accuracy and resolution. The calibration and validation of these quantities, for the R/Vs L'Atalante and Maria S. Merian, performed by either IFREMER or GEOMAR, are in agreement with international recommendations from GO-SHIP. There are two main differences between both methodologies of deployment: for IFREMER, the first measurements started at a depth of 5 meters, while for GEOMAR they started at around 1.5 meters. For IFREMER, the upward movement of the CTD package was stopped for 30 seconds before closing the Niskin bottles, while for GEOMAR they were closed underway. To assess any
discrepancies, a calibration of the raw measurements from the R/V L'Atalante CTD were performed with the GEOMAR procedures. Again, there are two notable differences in terms of calibration: First, the positioning of each CTD station was defined differently. IFREMER positions the CTD station taking the location and time at the start of the recording, whereas GEOMAR practice is to use the average position and time of the CTD station. Second, while the GEOMAR toolbox calibrates all the

profile together, IFREMER applies a piecewise calibration on sequences of 5 to 6 profiles at a time.


Figure 4 displays the difference between the two calibration methods applied to the R/V L'Atalante CTD profiles. For each parameter, most of the differences are found within the range of the sensor's accuracy, as provided by SeaBird. For temperature, differences range between -0.7 and 0.25 °C, with a mean difference of -3 $10^{-3}$ °C, and a median difference of -6 $10^{-4}$ °C. Figure 4d, we observe that the highest differences are situated between 50 and 250 m of depth, within the thermocline. For

salinity and oxygen, the distributions of the differences are found within the accuracy of the sensors, and median and mean difference are close to zero. However, Figure 4d shows that at the base of the mixing layer a salinity difference is sizeable and amounts up to 0.3 psu. These differences, albeit minimal, rise the question of CTD validation and calibration best practices. Both IFREMER and GEOMAR follow the recommendations from GO-SHIP in terms of CTD deployment and calibration. However, we observe that slight differences in the procedures can lead to non-negligible differences when the vertical gradi-

ents are more pronounced. Below these depths, the different approaches showed no major discrepancies, as the differences are inferior to the sensors' accuracy.

In general, at each isopycnal level, the differences between two profiles with small temporal and spatial separations can be linked to two sources: different calibration procedure and internal ocean variability. As the IFREMER and GEOMAR CTDs

calibration procedures provide similar results, around the same order of magnitude as the sensors' accuracy, the remaining differences between the various datasets must be due primarily to ocean variability. Figure 5a and 5b present the salinity absolute differences on isopycnal level, for each R/V, for CTD profiles found within a specific distance and time, respectively. As observed here, the averaged salinity differences, and associated standard deviation, depend on density. The largest variations are particularly noticeable on isopycnal levels, within which NBC rings evolve, as well as for denser water masses that were

sampled less frequently during the EUREC[4]A-OA field experiment. Moreover, as it was expected, these differences increase with time and distance (see Figure 5c and 5d for the vertically averaged absolute salinity differences), independently of the area of deployment. The R/V L'Atalante displays higher variability compared to the R/Vs Maria S. Merian and Meteor, mostly linked to the different areas observed. The variability more than doubles for CTD profiles distant more than 25 km or undertaken more than 4 hours apart. This illustrates the importance, for intercalibration purposes, to focus on close-by pairs of

profiles.

The calibrated CTD dataset for each R/V is associated to an uncertainty for every parameter. However, the creation of an assembled dataset gathering all profiles requires a comparison of these arrays. Figure 6 displays the vertically averaged differences for each parameter. The calculations are made on isopycnal levels. The left column shows these differences for profiles

performed by the same R/V as a function of both time and distance, also represented by the blue and red curves on the right panels. These curves exhibit the oceanic variability plus the post-calibration uncertainty; for near-by CTD pairs, they tend towards this last value. The central column of Figure 6 represents the difference calculated with one CTD profile from each R/V, synthesized by the green curves on the right panels. This curve is the result of both oceanic variability and uncertainty

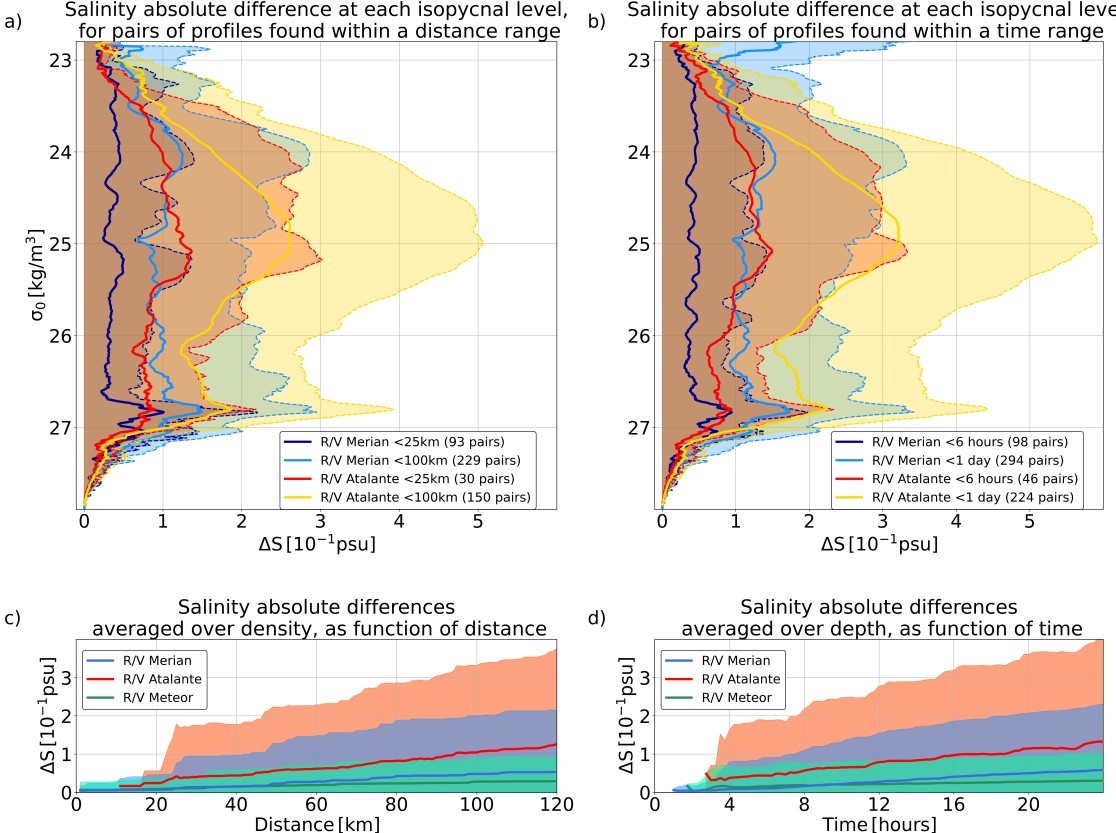

**Figure 5.** a) Absolute salinity difference between CTD pairs of profiles calculated on isopycnal levels for each R/V, averaged by distance. The dark blue and red curves are the average differences for profiles that are separated by less than 25 km for the R/Vs Maria S. Merian and L'Atalante respectively, while the blue and orange curves are for profiles found within 100 km. The shaded areas represented the standard deviation at each isopycnal level. b) Same as a) but with profiles averaged by time. The dark blue and red curves are for profiles separated by less than 6 hours, while the blue and orange are found within the same day, again for the R/Vs Maria S. Merian and L'Atalante respectively. c) Absolute salinity differences averaged vertically over the isopycnal levels as a function of distance for the R/Vs Maria S/ Merian (blue), L'Atalante (red), and Meteor (green). D) Same as c) but as a function of time instead of distance.

linked to the differences of deployment and calibration between each R/V and laboratory. For all the measured parameters,
these differences are found between those calculated for individual R/V, on average and standard variability. This indicates that the uncertainty remains low in terms of order of magnitude (equivalent to that provided by the laboratory calibration). The same comparison performed between the CTD profiles acquired by the R/Vs Maria S. Merian and Meteor are shown in Figure 7. They show small differences in temperature and salinity, but these remain within the standard deviation of the differences computed for profiles acquired by the individual R/V. This is expected since the CTD temperature and conductivity profiles undertaken by the R/V Meteor are calibrated using close-by CTD profiles acquired by the R/V Maria S. Merian. The resulting
uncertainties for the R/V Meteor temperature and salinity profiles are $2 \ 10^{-2}$ °C and $5 \ 10^{-3}$ psu respectively, and this with 3

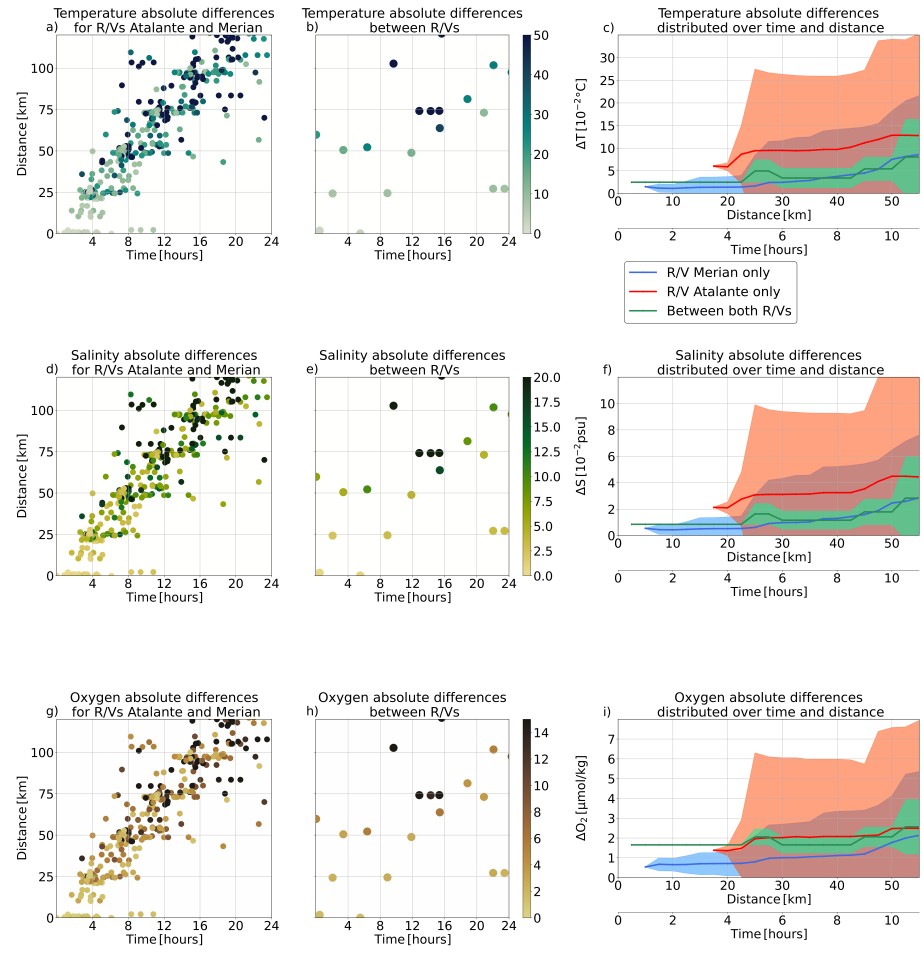

**Figure 6.** a) Absolute temperature differences between CTD pairs of profiles on isopycnal levels and averaged vertically distributed in time and distance. Each CTD pair is only composed of profiles from the R/Vs L'Atalante or Maria S. Merian. b) Same as a) but for CTD pairs composed of one profile from the R/V L'Atalante and one from the R/V Maria S. Merian. c) Temperature differences as a function of time and distance for CTD pairs from the R/Vs Maria S. Merian (blue), L'Atalante (red), and composed of one profile of each (green). d), e), and f) same as the above line but for absolute salinity differences, and g), h), and i) for absolute oxygen differences.

pairs of profiles found less than 5 km apart. Nevertheless, for dissolved oxygen this difference is of 4 $\mu$mol/kg, about the same order of magnitude of 5$\mu$mol/kg found for the uncertainty linked to the different calibration procedures.

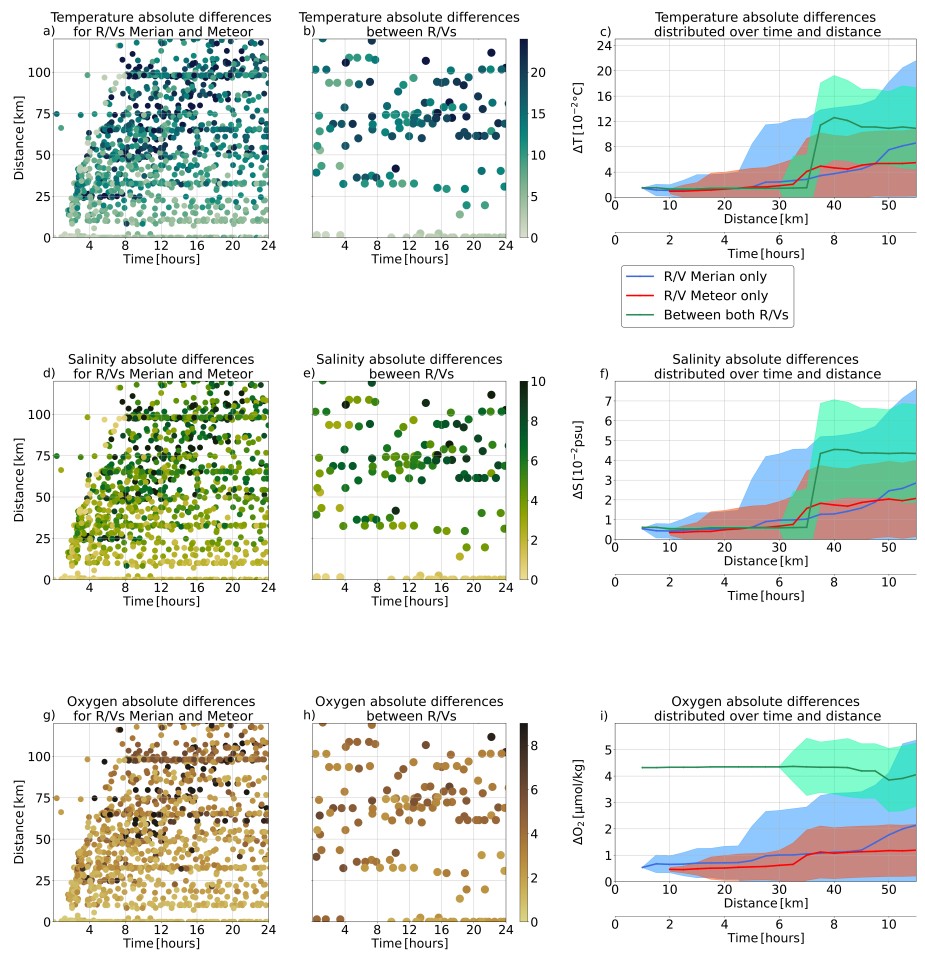

**Figure 7.** Same as Figure 6 but for the comparison between the R/Vs Maria S. Merian and Meteor.

## 3.3 Ship intake water analysis

### 3.3.1 Thermosalinograph

The R/Vs L'Atalante and Meteor are equipped with Seabird SBE21 TSG, the R/Vs Maria S. Merian and Meteor used two Seabird SBE45 TSG. These devices continuously measure temperature and salinity near the surface, between 6 and 7 m depth, depending on actual ship drought. The measurements are made at a frequency of 1 Hz and then averaged in 2-minute bins. Each day, for the R/Vs Maria S. Merian and L'Atalante, a water sample is taken and analyzed aboard in order to adjust the salinity measurements. Figure 8 shows the positions of the TSG records during the EUREC[4]A-OA experiment.

As the measurements from this device are compared and corrected with actual water samples measured with level 1 sensors on the R/Vs L'Atalante and Maria S. Merian, the calibrated TSG records are at level 2 of our calibration hierarchy. For the R/V

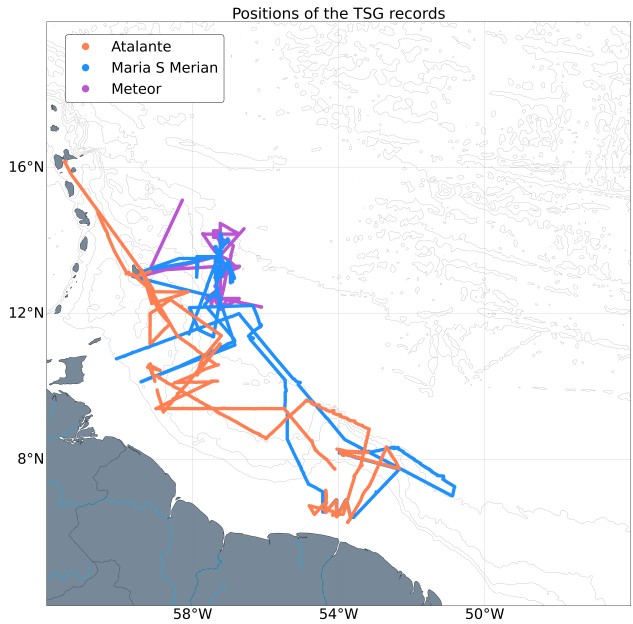

**Figure 8.** Map of the TSG records performed by the R/Vs L'Atalante, Maria S. Merian and Meteor.

Meteor, the level is set at 3 for the salinity measurements.

The absolute differences for surface temperature and salinity, calculated from the TSG, are displayed on Figure 9. Compared to the CTD measurements (see Figures 6 and 7), the TSG resolution allows for finer spatio-temporal scales of observations. The R/Vs L'Atalante and Maria S. Merian captured on average the same absolute differences for both temperature and salinity, while the R/V L'Atalante presents larger standard deviation. Nevertheless, the difference calculated between the two ships remains on average around $2\ 10^{-2}$ °C and $4\ 10^{-2}$ psu above the ones calculated for each individual R/V. Similar offsets are found for the R/V Meteor (not shown). For temperature, the standard deviation of the difference calculated between the TSG of each R/V remains in the same range as the ones obtained for individual TSG, but for salinity it exceeds the individual TSG's ranges. These differences can be attributed to the calibration processes, to the differences between the two TSG types, and to the background oceanic variability. Overall, the TSG assemble from all R/Vs is used for comparison with other devices measuring the surface with traceability of level 3 or below.

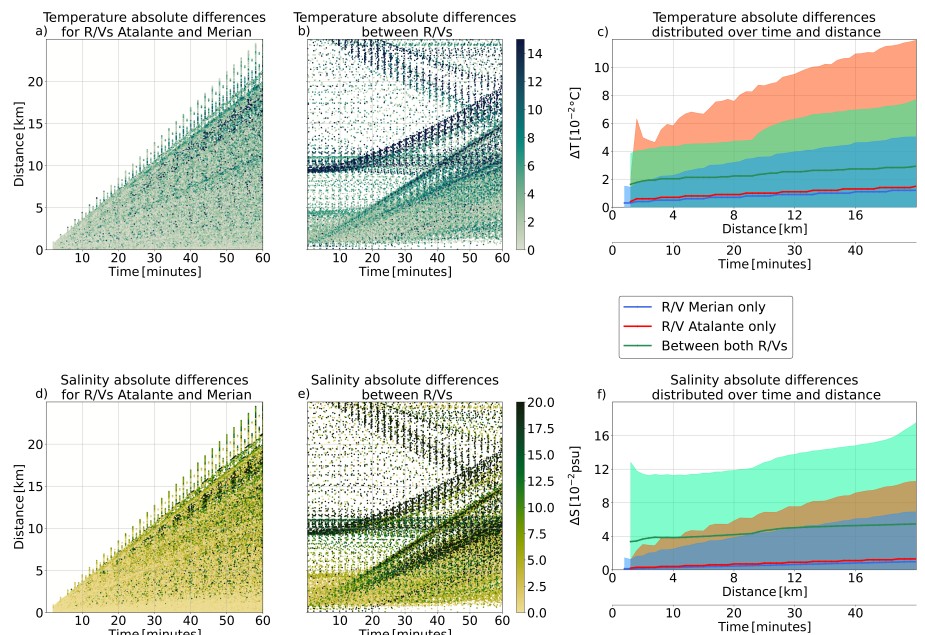

**Figure 9.** a) Absolute temperature differences between TSG pairs of measurements distributed in time and distance. Each TSG pair is only composed of measurements from the R/Vs L'Atalante or Maria S. Merian. b) Same as a) but for TSG pairs composed of one measurement from the R/V L'Atalante and one from R/V Maria S. Merian. c) Temperature differences as a function of time and distance for CTD pairs from the R/Vs Maria S. Merian (blue), L'Atalante (red), and composed of one profile of each (green). d), e), and f) same as the above line but for absolute salinity differences.

## 3.4 Underway platforms

### 3.4.1 Underway CTD (uCTD)

An Ocean Science underway CTD system was used on the R/Vs L'Atalante and Maria S. Merian. The uCTD consists of a small winch system mounted on the bulwark of the ship and a CTD probe measuring temperature, conductivity, and pressure (Rudnick and Klinke, 2007). The probes sample at 16 Hz and data is recorded internally and read out onboard via a Bluetooth connection. This probe is designed to record data during descent, and the deployment procedure minimizes the influence of surface waves. The uCTD can be used in "free-cast" and "tow-yo" (or free fall) modes, depending on whether a winding line

is either spooled or not onto the tail. In the first case, the probe descent is decoupled from the winch spool friction, keeping a rather steady descent rate of about 4 dbar/s. In the second mode, the probe fall rate is determined by the idle friction of the spool and the probe friction, and descent may vary between 3.5 dbar/s and about 0.6 dbar/s. For deployments of the uCTD from the R/V Maria S. Merian, both modes were used, with the "tow-yo" preferred for water depths shallower than 500 meters. For deployments from the R/V L'Atalante, the uCTD was only deployed using the free fall mode, with no line spooled onto

the tail. The difference in descent rate is an important factor to take into account in the calibration procedure, as the water is

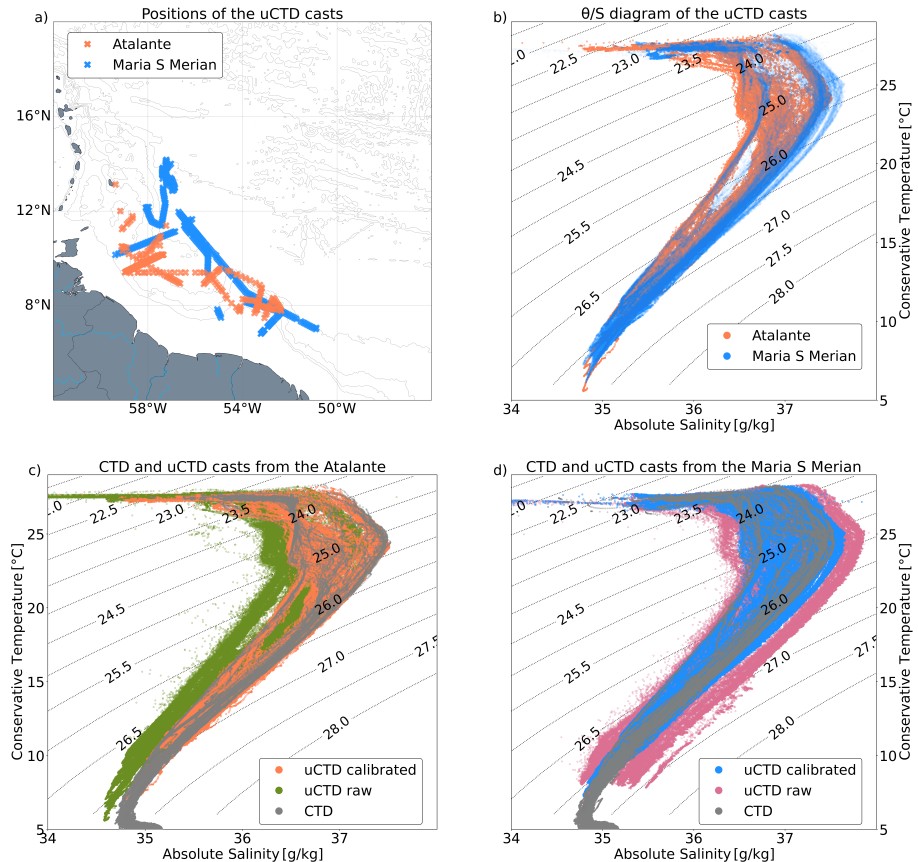

**Figure 10.** a) Map of the uCTD casts positions for the R/Vs L'Atalante and Maria S. Merian. b) θ/S diagram of the uCTD calibrated profiles for each ship superimposed on the isopycnals. c) and d) Comparison of the raw and calibrated uCTD profiles superimposed on the CTD profiles for R/Vs L'Atalante and Maria S. Merian respectively.

not pumped toward the sensors, contrary to what happens for sensors mounted on the CTD.

The R/V Maria S. Merian acquired a total of 380 profiles using three different uCTD probes, usually performed in series with a drop every 30 minutes. The R/V L'Atalante carried out 179 uCTD profiles using two probes and usually alternating

their deployment with CTD profiles. Figure 10a shows the positions of these profiles. Depending on the ship speed, the uCTDs sampled the water column between the surface and 300 to 500 meters of depth. Because no real-time information on the actual depth of the probe is available, the operator has to estimate the probe depth via cast time and estimated sinking velocity.

There are four sources of salinity uncertainty associated with the uCTD measurements leading to errors of about 4–5 $10^{-3}$

°C, resulting in a computed salinity error of 4–5 $10^{-3}$ psu (Ullman and Hebert, 2014). The first arises from a looping of the probe during which its direction reverses. The uCTD is made for recording data only during descent, so periods of inverse

descent rate can be easily removed during the validation procedure. The second source of error comes from the variable fall rate of the probe. As the water is not pumped towards the sensors, there can be a lag in the measurements linked to the time it takes for the parcel of water to pass from one sensor to the other (Perkin et al., 1982; Lueck, 1990). Corrections are achieved via
a minimization of this temporal lag, which depends on the fall rate. The third source of error is related to the high deceleration of the probe when the spooled line reaches its end, leading to viscous heating of the thermistor. Larson and Pedersen (1996) provides a correction for this effect that is taken into account before computing salinity. The final source of error for salinity is detected as staggered changes when comparing the CTD profiles to those of the uCTD. Indeed, salinity values are lower for the uCTD than for CTD profiles when the temperature increases. This conductivity cell thermal mass error has been described in
Lueck (1990), and Lueck and Picklo (1990) proposed a correction based on the calculation of two parameters: the magnitude of the error $\alpha$ and a time constant of the error $\tau$.

To perform the thermal mass correction, it is necessary to have close-by CTD/uCTD profiles. Specific profiles were also undertaken with the uCTD probe directly attached to the CTD rosette to gather the most synoptic measurements possible.
Nevertheless, while the uCTD is designed to record the water column in a free fall mode, with the probe attached it is lowered at a constant but lower speed, and the presence of the rosette can disturb the flow near the sensor intake. As a consequence, even the co-located uCTD-CTD profiles can underestimate the error.

All corrections are performed using a Matlab toolbox based on Ullman and Hebert (2014). The procedure uses two separate
calibrations: direct calibration of the uCTD probe attached to a CTD, giving nearly collocated uCTD and CTD profiles, and comparison with the TSG salinity. Figure 10b shows the calibrated profiles, and Figure 10c and d present their comparisons with CTD and raw uCTD profiles for each ship.

For the corrected uCTD, some calibration profiles were performed with the uCTD probe attached to the CTD rosette, while
others were calibrated by comparing to nearby CTD stations. Both these methods rely on comparing the uCTD profiles with close-by and sometimes synoptic CTD profiles, and are thus assign at level 3 of our calibration hierarchy. After the correction of the uCTD data, we observe a clear improvement and agreement between the CTD and uCTD profiles. Figure 11 shows the vertically averaged differences between CTD and uCTD measurements for temperature and salinity for each individual R/V. Since the uCTD is calibrated with nearby CTD profiles it is particularly interesting to notice that the correction highly reduces
the differences in both temperature and salinity, for the mean and standard deviation, leading them closer to the differences as measured by only the CTD. For the R/V L'Atalante the comparison matches better than for the R/V Maria S. Merian, possibly linked to the deployment strategy, where uCTD and CTD measurements were alternately cast for the R/V L'Atalante, increasing the number of profiles to use as references for calibrations, while on the R/V Maria S. Merian full sections only used uCTD casts. The difference between the CTD only curve and the one combining CTD and uCTD provides an estimation
of the uncertainty of 9 $10^{-2}$ °C and 2 $10^{-2}$ psu for both ships, with 5 pairs of profiles found within 10 km and 2 hours for the

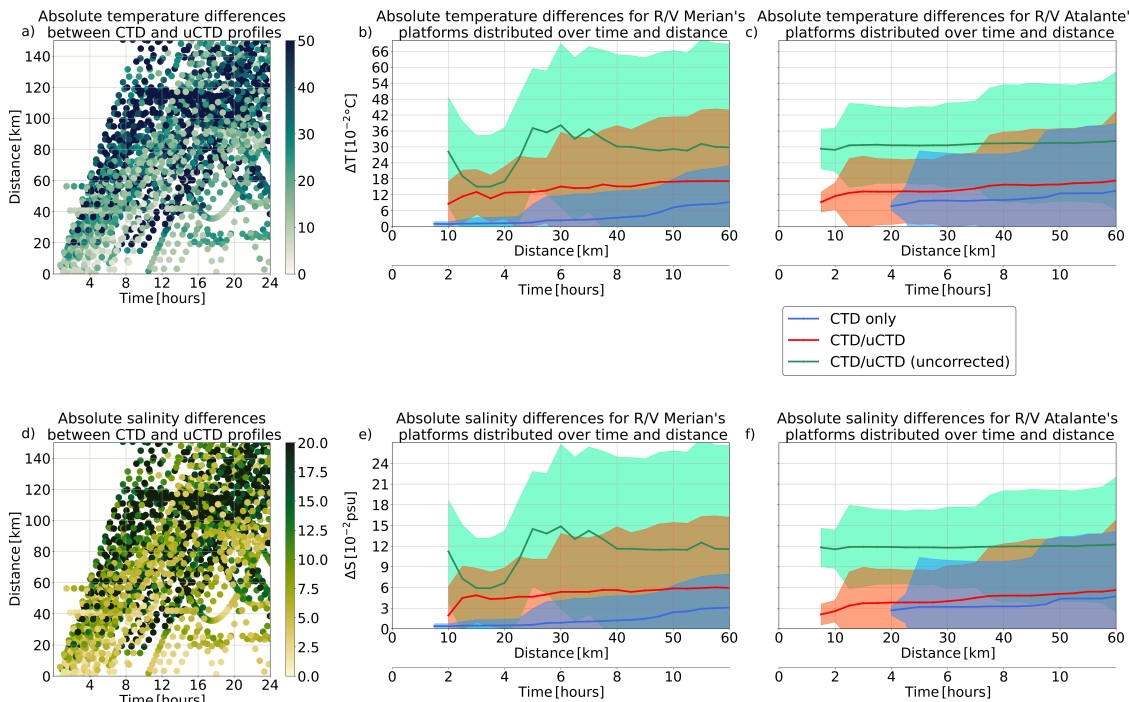

**Figure 11.** a) Absolute temperature differences between CTD/uCTD pairs of profiles on isopycnal levels and averaged vertically distributed in time and distance. Each pair is composed of profiles from the same R/V, one CTD and one uCTD. b) Temperature differences as a function of time and distance for pairs of profiles from the R/V Maria S. Merian. In blue for pairs of CTD only profiles, in red for pairs composed of one CTD and one uCTD profiles, and in green for pairs composed of one CTD and one uncalibrated uCTD profiles. c) Same as b) but for the R/V L'Atalante. d), e), and f) same as the above line but for absolute salinity differences.

R/V Maria S. Merian and 10 pairs for the R/V L'Atalante.

### 3.4.2 Moving Vessel Profiler (MVP)

A Moving Vessel Profiler (MVP) 30-350 from AML was operated on the R/Vs L'Atalante and Maria S. Merian. The MVP
allows underwater measurements from the surface down to a depth that depends highly on the ship and water current speed, reaching a maximum of 350 meters. The MVP 30-350 consists of an electric winch system, a PC control unit and the towed vehicle ("fish") that can be equipped with different sensors. A conductive probe provides real-time data access. The "fish" is forced to rapidly (up to 2 m/sec) descend and ascend with the help of a tail-unit. The fish design enables descent, ascent, and drag phase (time before the next descent) data recordings to be used.


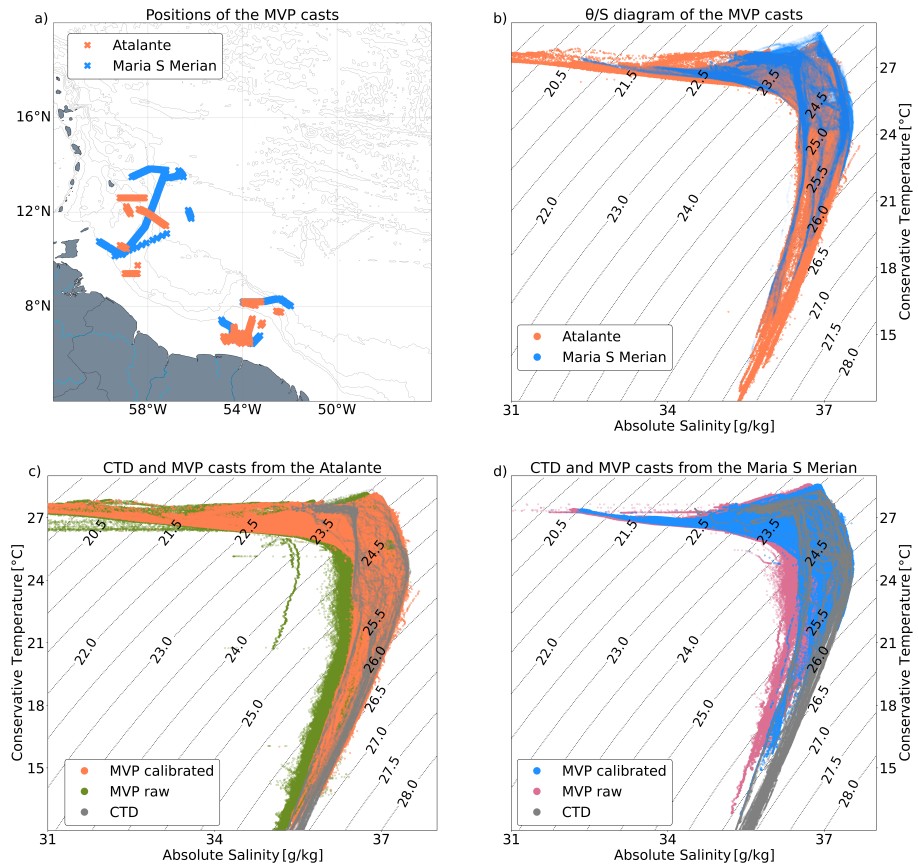

**Figure 12.** a) Map of the MVP casts positions for the R/Vs L'Atalante and Maria S. Merian. b) $\theta$/S diagram of the MVP calibrated profiles for each ship superimposed on the isopycnals. c) and d) Comparison of the raw and calibrated MVP profiles superimposed on the CTD profiles for the R/Vs L'Atalante and Maria S. Merian respectively.

The MVP deployed from the R/V Maria S. Merian performed 1891 profiles with ship speed from 2 to 10 kn (mean 6.9 kn). The MVP deployed from the R/V L'Atalante completed 1960 profiles (Figure 12a). The device was operated in different areas and guided by dynamical features (mesoscale eddies, filaments, frontal regions). In regions with a shallow topography, the maximal diving depth was either controlled by an operator (real-time data) or by feeding the bottom topography (e.g., from the ships' echo sounder) into the MVP control unit (as done on Maria S. Merian in shallow topography and requesting the fish to ascend when reaching a depth of 10 m above the seafloor). On the R/V Maria S. Merian the MVP was also equipped with a fluorescence sensor.

The CTD sensors on the MVP are affected by similar sources of error as the uCTD: the speed of the probe through the water, CT time-lag, and thermal mass, and a similar calibration strategy is therefore used for the MVP sensors. The MVP also appears to be sensitive to surface waves. We therefore removed them by applying a low-pass filter with a cut-off frequency

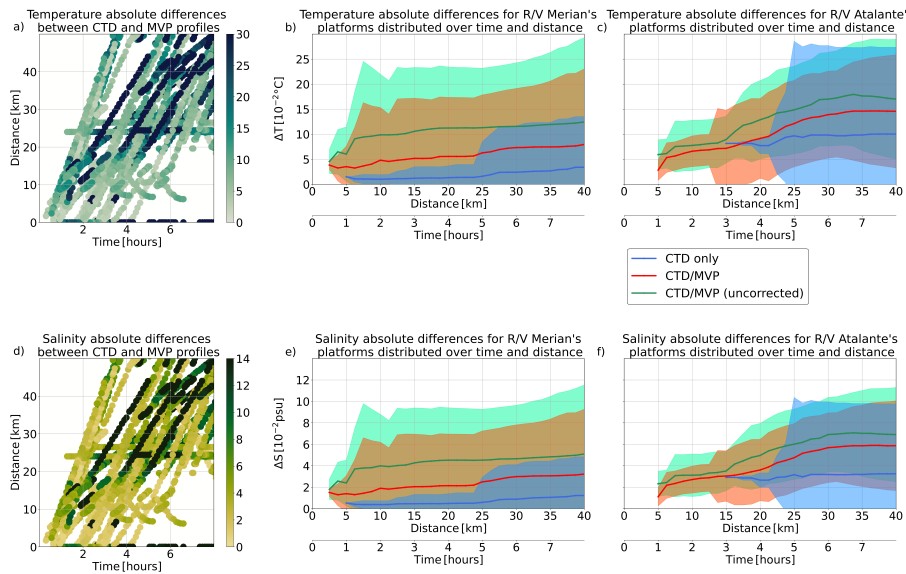

**Figure 13.** a) Absolute temperature differences between CTD/MVP pairs of profiles on isopycnal levels and averaged vertically distributed in time and distance. Each pair is composed of profiles from the same R/V, one CTD and one MVP. b) Temperature differences as a function of time and distance for pairs of profiles from the R/V Maria S. Merian. In blue for pairs of CTD only profiles, in red for pairs composed of one CTD and one MVP profiles, and in green for pairs composed of one CTD and one uncalibrated MVP profiles. c) Same as b) but for the R/V L'Atalante. d), e), and f) same as the above line but for absolute salinity differences.

calculated from the surface waves of 0.2 Hz. As explained in the uCTD section, the temporal lag and thermal mass errors are linked respectively to the vertical speed and the time match between conductivity and temperature pairs. In order to perform this calibration, we calculated downward and upward correction coefficients based on the method from Mensah et al. (2018). The calibration resulted in a good match between nearby ship CTD and CTD sensors on the MVP (Figure 12c, d).

Because the calibration takes into account only nearby CTD profiles, temperature, salinity, and pressure measurements are ranked level 3 of traceability. We observe (Figure 13) that the averaged differences between nearby CTD profiles, and standard deviations reduce after calibration. As for the uCTD comparisons with the CTD, a difference between CTD only and MVP/CTD ones subsist. Again, this is attributed to the different devices' deployment, the sampling strategy and the oceanic variability of the regions measured. The estimated uncertainty for the corrected MVP is of $3 \ 10^{-2}$ °C, and $1 \ 10^{-2}$ psu for the R/V L'Atalante, and $4 \ 10^{-2}$ °C, and $2 \ 10^{-2}$ psu for the R/V Maria S. Merian, with respectively 10 and 70 pairs of profiles found within 5 km and 1 hour.

## 3.5 Ship mounted ADCP (S-ADCP)

Upper ocean currents were measured quasi-continuously with Teledyne RD Instruments Ocean Surveyor Acoustic Doppler Current profiler (S-ADCP). On all three ships a 38 kHz ADCP was operated, measuring velocities from around 50 to sometimes even below 1000 meters depth depending on the availability of scatters. The R/Vs Meteor and Maria S. Merian were also equipped with a 75 kHz ADCP, providing measurements between 40 and 800 meters of depth with a finer resolution compared to the 38 kHz S-ADCP. The R/V L'Atalante, on the other hand, was equipped with a 150 kHz ADCP, supposedly ranging from around 20 meters to 400 meters depth. However, this ADCP rarely reached water depths below 200 meters during the experiment. The ADCP accuracy is $\pm 5\%$ of measured velocity or $\pm 0.5$ cm/s, whichever is greater.

During the experiment, on a daily basis, data from the R/V L'Atalante were then processed with the CASCADE software developed by IFREMER (Le Bot et al., 2011; Speich et al., 2021b). The data from the R/Vs Maria S. Merian and Meteor was processed using a set of Matlab routines applied to the raw data, meaning the data was sampled in an as-fast-as-possible mode (approximately 1Hz).

To enable a direct comparison of the measurements from all ships, the ADCP data was reprocessed using the UHDAS routines Firing et al. (2012). This comparison did not show any notable difference among the data sets. The final data set consists of zonal and meridional velocities averaged in two-minutes segments. As the device is directly mounted on the hull of the ships, the measurements follow the same tracks as the TSG (Figure 8).

By default, we place the processed S-ADCP measurements on the second level of our hierarchy, since only an intercomparison between devices can be done and no comparison with a reference is possible. The L-ADCP measurements are placed one level below (level 3), since S-ADCP measurements are used as reference.

# 4 Autonomous devices

## 4.1 Underwater Gliders

### 4.1.1 Pressure, Temperature, Conductivity, and Salinity

During the experiment, 7 underwater gliders were deployed (Figure 14). From the R/V L'Atalante a SeaExplorer Glider, named Kraken, was deployed and carried out 831 profiles to a maximum depth of 700 meters in 15 days of operation. The glider was deployed to cross along different quadrants of a mesoscale eddy previously located in satellite altimetry maps. Its CTD, a Glider Payload CTD (GPCTD), has an accuracy of $4 \ 10^{-3}$ °C for temperature and of $1 \ 10^{-3}$ mS/cm for conductivity claimed by the manufacturer.

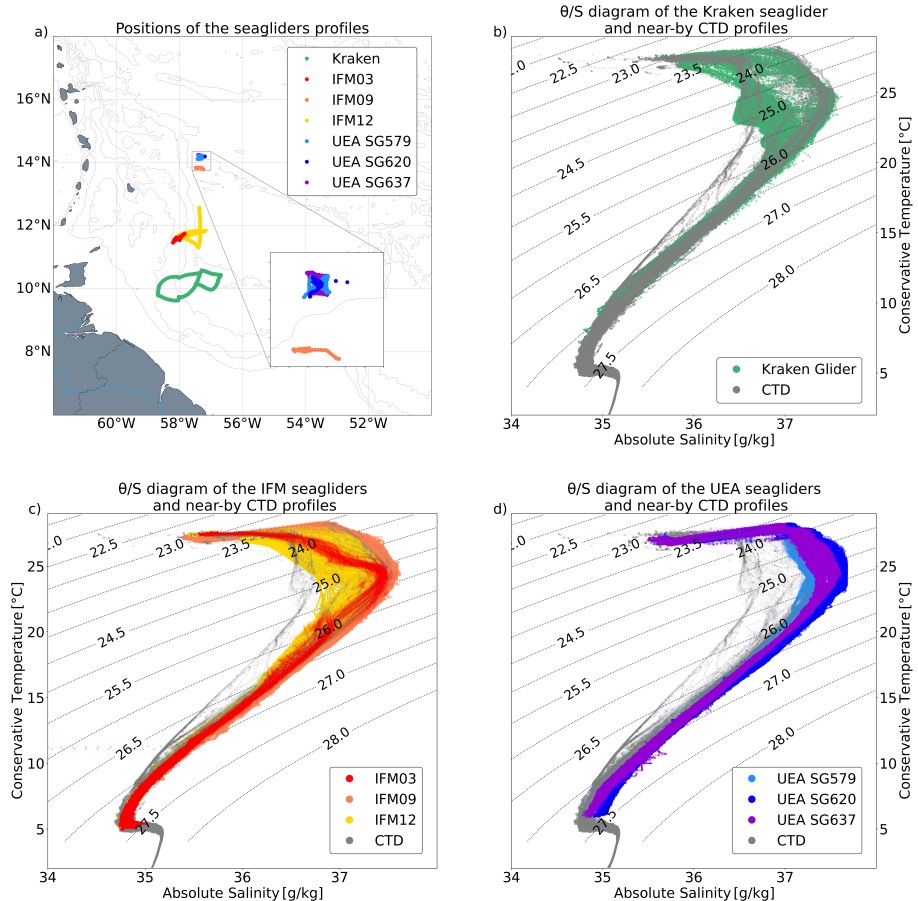

**Figure 14.** a) Map of the 7 underwater gliders positions. b) $\theta$/S diagram of the Kraken glider measurements, deployed from the R/V L'Atalante, compared to near-by CTD casts. c) Same as b) for the IFM gliders deployed from the R/V Maria S. Merian. d) Same as b) for the University of East Anglia gliders deployed from the R/V Meteor.

From the R/V Maria S. Merian, 3 autonomous gliders (IFM03, IFM09, IFM12) have been deployed. IFM09 conducted 327 profiles to a maximum depth of 900 meters in 20 days following a quasi-stationary mode in the trade-wind alley area. IFM03 and IFM12 were deployed in the northeastern region of a mesoscale eddy, performing respectively 125 and 443 profiles down to a maximum depth of 900 meters in 6 and 24 days, respectively. Due to a leak, IFM03 was retrieved by the R/V Meteor.


Three gliders from the University of East Anglia (UEA) collected profiles with an "hourglass" sampling pattern. The gliders (SG579, SG620, SG641) collected 442, 262 and 308 profiles for 10, 13 and 24 days down to maximum depths of 950, 750 and 750 meters, respectively.

The IFM and UEA gliders were equipped with unpumped SBE41 Seabird CTDs. (Stevens et al., 2021), with manufacturer accuracy of $2 \cdot 10^{-3}$ °C for temperature and of $3 \cdot 10^{-3}$ mS/cm for conductivity. The underwater gliders are subject to the same sources of errors as all the previously described undulating probes. Nevertheless, the rising and descending profiles are performed at a much lower speed compared to the free-fall of the uCTD. For Kraken the water was pumped through the CTD and this controlled flow highly diminished the viscous heating and temporal lag effects. For the other underwater gliders this was

not the case and the flow speed was estimated by determining the speed of movement through the water based on an optimized glider travel model.

In principle, similar correction procedures were used for the MVP and underwater gliders. These were necessary to avoid any misalignment between the downward and upward profiles, taking into account the flight model, and to have coherent cali-

brations across the platforms.

No direct or lab calibration was done with the glider CTD sensors; instead, data calibration was performed through comparison with nearby (time, space) CTD stations. To ease comparison we group all three UEA Seaglider, the IFM gliders, and the Kraken (Figure 15 above and middle line panels). The number of close-by pairs of CTD and glider profiles is quite low for the

Kraken. Nevertheless, the mean differences for pairs of profiles found closer than 15 km (with 15 pairs of profiles) are less than $3 \cdot 10^{-2}$ °C, and $1 \cdot 10^{-2}$ psu. The IFM gliders tend to constant positive bias in temperature and salinity compared to the near-by CTD stations, with a low standard deviation despite having a higher number of pairs of profiles. Within the first 10 km (about 60 pairs of profiles) the uncertainty is on the order of 0.1 °C, and $4 \cdot 10^{-2}$ psu. The UEA Seaglider shows higher differences in temperature and salinity associated with a large standard deviation. These values can be attributed to the calibration and

differences between devices, but also to the sampling strategy; the pairs UEA Seaglider/CTD are mostly associated to the R/V Meteor stations (more than 140 pairs of profiles within the first 10 km) (see Figure 3a) where the salinity has not been calibrated with a direct comparison with water samples. This probably impacts the resulting isopycnal levels where the absolute difference is calculated. Their uncertainty is here estimated at $1.8 \cdot 10^{-1}$ °C and $7 \cdot 10^{-2}$ psu. As for the MVP measurements, the underwater gliders compared to the CTD profiles from the R/Vs Maria S. Merian and L'Atalante are placed at level 3 on

the calibration hierarchy. The UEA gliders are principally compared with CTD profiles acquired by the R/V Meteor and thus placed one level below.

### 4.1.2  Dissolved Oxygen

The three IFM gliders were equipped with AADI Aanderaa optodes of type 3830 (IFM03, IFM09) and type 4831 (IFM12).

The manufacture-provided resolution and accuracy for oxygen concentration are 1 mmol and 8M or 5% (whichever is greater, concentration) and < 5% (air saturation). For the three IFM gliders, lab calibrations of the oxygen sensors were done on board the Maria S. Merian by preparing zero (chemically forced) and 100% saturation (air bubbles injected) water of two

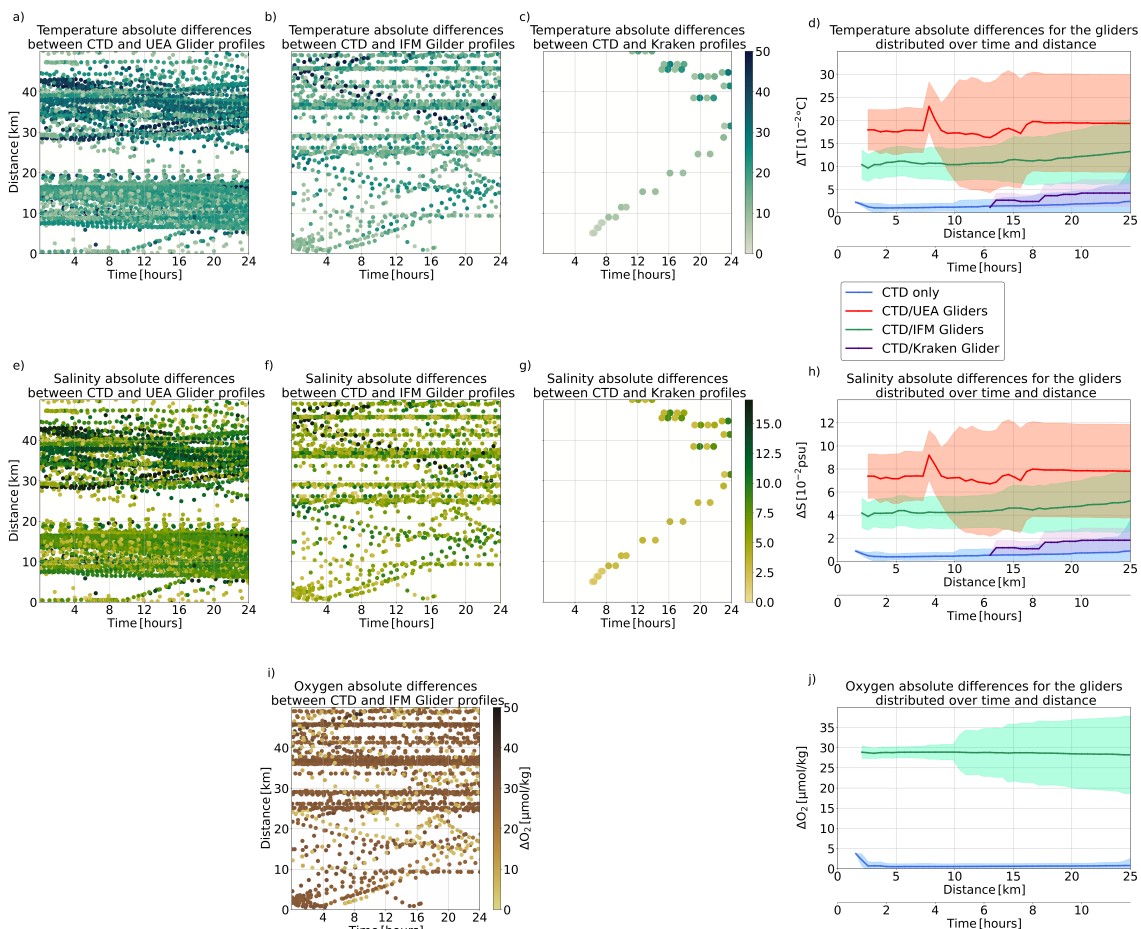

**Figure 15.** a), b), and c) Temperature absolute differences between CTD/Glider pairs of profiles on isopycnal levels and averaged vertically distributed in time and distance. Each pair is composed of one CTD and one Glider profiles. Respectively a), b), and c), correspond to the UEA, IFM, and Kraken Gliders. d) Temperature differences as a function of time and distance for pairs of profiles. In blue for pairs of CTD only profiles, in red for pairs composed of one CTD and one UEA profiles, in green for pairs composed of one CTD and one IFM profiles, and in purple for pairs composed of one CTD and one Kraken profiles. The middle line of panels e) to h) is the same as above, but for salinity absolute differences. The bottom line of panels correspond to oxygen absolute differences, but only IFM Gliders measured dissolved oxygen.

temperatures, following the Aanderaa optode manual. The resultant readings were used to constrain the phase / temperature relation of the foil.

Figure 15j provides information about oxygen concentration with, on average, negative (not shown) differences of $28\mu$mol/kg. However, this relatively high uncertainty value has to be put in perspective with the associated standard deviation of about $\pm 2\mu$mol/kg for CTD/Glider pairs found within 10 km. This uncertainty has the same order of magnitude as the sensor accuracy, suggesting a rather constant bias in the sensors. The oxygen as measured by the IFM gliders is placed at level 3 of the

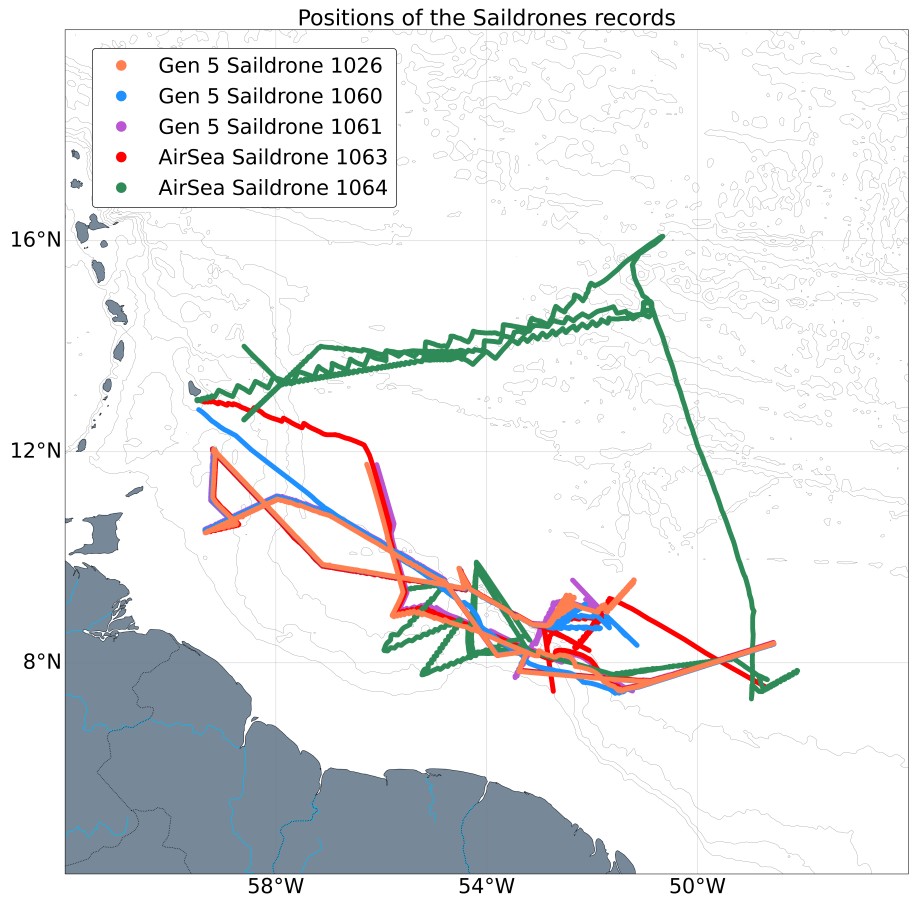

**Figure 16.** Map of the five Saildrones positions. Three NASA Saildrones (1026, 1060 and 1061) and one NOAA Saildrone (1063) have followed the R/Vs Maria S. Merian and L'Atalante tracks to sample surface mesoscale eddies, while one NOAA Saildrone (1064) has first sampled the trade wind alley.

calibration hierarchy.


### 4.1.3 Other sensors

CDOM optical sensors were mounted on the IFM12, SG579, and Kraken gliders to estimate dissolved organic matter. A SUNA nutrient analyzer was used on IFM12. IFM03 and SG620 were equipped with a Rockland Scientific MicroRider turbulence sensor to estimate small-scale mixing. The SG637 glider was equipped with a Nortek Signature1000 1 MHz ADCP to measure
the vertical shear of horizontal currents. None of these sensors is considered here.

## 4.2 Saildrones

With the objective of measuring the ocean-atmosphere interface, five Saildrones were deployed from Barbados, three funded by NASA and two by NOAA. Below the water line, a pumped CTD (SBE-37-SMP-ODO Microcat) measured temperature, conductivity, and dissolved oxygen at 0.5 meter depth, and they were also equipped with a chlorophyll-a sensor (Wetlabs ECO-FL-S G4 and Turner Cyclops). A Teledyne Workhorse 300kHz ADCP was mounted on the NASA Saildrones to measure the current velocity from 6 to 100 meters depth. Their nominal accuracies are $\pm$ 2 $10^{-3}$ °C for temperature, $\pm$ 3 $10^{-3}$ mS/cm for conductivity, $\pm 3$ $\mu$mol/kg or $\pm 2\%$ for dissolved oxygen, and 2 $10^{-2}$ $\mu$g/l for chlorophyll-a. These values have the same order of magnitude as those for the sensors mounted on the CTD probes. The ADCP accuracy is $\pm 5\%$ of measured velocity or $\pm 0.5$ cm/s, similar to that of the S-ADCPs. The Saildrones are autonomous and operated remotely, and their measurements are valuable as they provide 1-minute averaged records of temperature and salinity near the sea surface, and 5-minute averaged records of velocity with high vertical resolution in the upper layer of the water column. Nevertheless, calibrations were only made in laboratory before and after the mission, as direct comparisons with water samples are impossible; as such, we place the Saildrones on level 3 of our calibration hierarchy.

As for the previous devices, Figure 17 shows the comparison between the temperature and salinity measurements made by the Saildrones and the near-surface values from the R/Vs L'Atalante and Maria S. Merian measurements. The Saildrones followed their own routes away from the ships, nevertheless the high frequency of acquisition provides enough nearby measurements for comparison. For calibration purposes, 4 of the 5 Saildrones spent times twice near the R/V L'Atalante. For temperature, measurements found below 1 km apart, more than 350 pairs of measurements were less than 1 km apart. The resulting difference is of the order of 2.5 $10^{-2}$ °C. For salinity, the absolute difference climbs to 4 $10^{-2}$ psu for measurements found less than 1 km apart. This difference reduces to 1 $10^{-2}$ psu when using the R/Vs TSG tendency.

## 4.3 Argo floats

During the experiment, 5 Argo floats were deployed from the R/V L'Atalante. As seen in Figure 18, only a very limited number of profiling floats drifted in the region in late 2019 and early 2020. Two of the deployed Argo floats were configured to follow a large surface-intensified mesoscale anticyclone, one was deployed in a region between two mesoscale eddies, and the two within a subsurface anticyclone. The positions of the deployments were determined from analyses of satellite altimetry maps and the detection of eddies with the TOEddies algorithm (Laxenaire et al., 2018), as well as with the analyses of S-ADCP data from the different ships. The floats were initially set to perform daily vertical profiles between 0 and 1000 meters depth, and when not profiling they were positioned to a parking depth at 200 dbar. After a period varying between 10 and 90 days, they were programmed to the core Argo setting (a parking depth of 1000 dbar and vertical profiles every 10 days between 2000 dbar and the surface). Their trajectories and vertical profiles were collected and validated by the Coriolis Argo Global Data Assembly Center (GDAC) in Argo (2000). These 5 floats are PROVOR manufactured by NKE. They were equipped with Seabird CTD

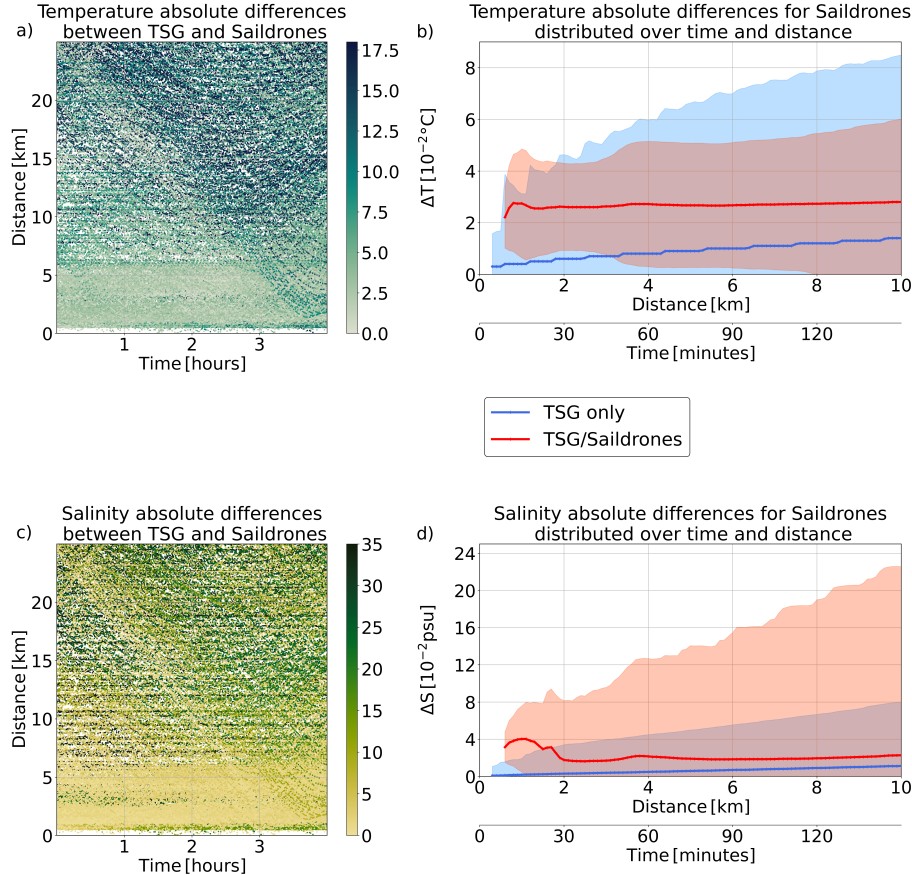

**Figure 17.** a) Absolute temperature differences between TSG/Saildrone pairs of measurements distributed in time and distance. Each pair is composed of one R/V's TSG measurement and a Saildrone's one. b) Temperature differences as a function of time and distance for a pair of surface measurements. In blue for pairs of R/Vs' TSG only profiles, in red for pairs composed of one R/V's TSG and one Saildrone's. c) and d), same as the above line, but for absolute salinity differences.

sensors SBE41, and their positions were transmitted via the Iridium system. They measured pressure, temperature, and salinity with respective accuracy of $\pm$ 2.4 dbar, $\pm$ 2 $10^{-3}$ °C, and $\pm$ 3 $10^{-3}$ psu. Each of the sensors may drift over the years, but any drift remained small over the period of the experiment.

As Figure 19 underlines, direct comparisons between Argo floats and nearby CTD stations are limited because of the low number of measurements. Moreover, since the float trajectories differ highly from that of the ship, the background oceanic variability estimated from the CTD might differ. Nevertheless, for nearby pairs of Argo and CTD profiles, the difference remains small. For ARGO floats and CTD stations found within 25 km, with 5 pairs of CTD/Argo profiles, the uncertainties for temperature and salinity are of 4 $10^{-2}$ °C and 1.5 $10^{-2}$ psu. These uncertainties are of the same order of magnitude as those

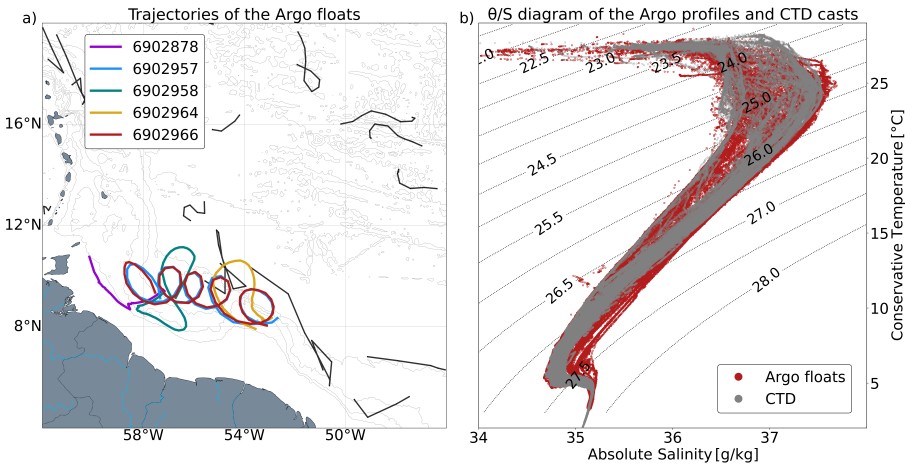

**Figure 18.** a) Map of the Argo floats found in the region of interest between December 2019 and May 2020; specific colors are attributed to the five floats deployed from the R/V L'Atalante. b) θ/S diagram of the Argo floats, compared to near-by CTD casts, no clear bias is observed here.

quantified for the other level 3 devices sampling the water column (uCTD and MVP).

Additionally, an OPTODE sensor (for dissolved oxygen) from AANDERAA with an accuracy of 4 $\mu$mol/kg was mounted on the Argo floats deployed from R/V L'Atalante. In the same spatial and temporal range as the CTD stations, the dissolved oxygen sensors exhibit an uncertainty of 10 $\mu$mol/kg for a 25 km range (see Figure 19f).

### 4.4 Surface Drifters

Four kinds of surface drifters were deployed from the R/V L'Atalante during the experiment. Two SURPACT drifters were launched for short periods of time, less than two days, and we will not describe them in detail here. Five SVP-BRST drifters from Eumetsat grant TRUSTED to MétéoFrance/CLS, measured temperature at 0.15 meters depth, with an accuracy of $\pm 5$ $10^{-3}$ °C. Two SVP-BSC drifters from MétéoFrance/LOCEAN with CNES SMOS support measured temperature and salinity at 0.2 meters depth, with respective accuracy of $\pm 0.1$ °C and $\pm 5$ $10^{-2}$ psu, configured to send data every 6 minutes with a sampling rate of 20 seconds. Finally, ten SVP-BSW drifters from NOAA were deployed to measure temperature and salinity, at 0.5, 5 and, 10 meters of depth with the same accuracy as SVP-BSC drifters, transmitting data every 30 minutes. Figure 20 shows the positions of these drifters, exhibiting clear advection toward the Northwest, with some of them looping inside mesoscale eddies.

From the R/V L'Atalante, two SVP-BSC and Surpact were deployed for short periods of time to compare their measurements with the similar devices and nearby instruments (Reverdin et al., 2021). Figure 21, similar to that for Saildrones, shows the comparison between surface TSG measurements and nearby drifter records. As for the Saildrones, these comparisons are

delicate because the drifters measured temperature and salinity at different vertical levels near the surface, capturing a sensibly different background variability. The observed difference for the SVP-BSW drifters remains small for near-by pairs of measurements (less than 5 km apart and more than 300 pairs of measurements), with an uncertainty of $2.5\ 10^{-2}$ °C for temperature and $2.5\ 10^{-2}$ psu for salinity. The SVP-BSC drifters' measurements exhibit a large difference in both temperature and salinity, $4.5\ 10^{-2}$ °C and $2\ 10^{-1}$ psu, with more than 250 pairs of measurements found within 5 km. These large differences are particularly linked to one drifter. This is related to a large thermal effect upon deployment, supposedly becoming near negligible after a few hours. The manufacturer states that we should not consider the first few hours of data. Finally, temperature measurements from the SVP-BRST drifters follow the uncertainty from the TSG only, with an offset of about $2\ 10^{-2}$ °C for 17 pairs of measurements found within 5 km.

## 5 Data Concatenation

All the measured and cross-validated parameters can be associated with a level of uncertainty while using a concatenated dataset. Table 1 summarizes all the uncertainties calculated in this study. For example, for a vertical section of salinity from the R/V L'Atalante using CTD, uCTD, and MVP measurements, the associated uncertainties are $3\ 10^{-3}$ psu for the CTD, $2\ 10^{-2}$ psu for the uCTD, and $1\ 10^{-2}$ psu for the MVP. Thus, the dataset provides an array with different uncertainties for each type of profile.

The concatenated data are very useful for enhancing the space-time resolution of many of the sampled areas and for better assessing dynamical properties of the regional ocean circulation. With these data it is, for example, possible to assess relevant properties of surface and subsurface mesoscale eddies, freshwater filaments, cold water pools and freshwater-induced barrier layers. Figure 22 shows the impact of the calibration on a specific section performed by the R/V L'Atalante using CTD, uCTD, and MVP measurements. The first column, displaying only the CTD profiles, has a coarse horizontal resolution but is associated with low uncertainty linked to the validation and calibration processes. The second column presents the complete section, with the uncalibrated MVP and uCTD profiles. There we observe an increase in the horizontal resolution but nevertheless find some inconsistencies, as seen in panels b and c with large variations between two successive profiles of salinity and potential density. Finally, after calibration, as seen in the last column, the uCTD and MVP profiles are corrected and cross-validated with the CTD measurements. The inconsistencies are removed while still keeping high horizontal resolution and low uncertainty, enabling the calculation of realistic gradients of the different fields and the analyses of derived parameters, such as the Ertel Potential Vorticity (see Figure 22e).

| Vertical Profiles | | | |
|---|---|---|---|
| | **Parameter uncertainty / Level of traceability** | | |
| **Platforms** | **Temperature** | **Salinity** | **Oxygen** |
| R/V L'Atalante CTD | $1\ 10^{-3}$ °C / Level 2 | $3\ 10^{-3}$ psu / Level 2 | $1.60\ \mu$mol/kg / Level 2 |
| R/V Maria S. Merian CTD | $1\ 10^{-3}$ °C / Level 2 | $2\ 10^{-3}$ psu / Level 2 | $0.61\ \mu$mol/kg / Level 2 |
| R/V Meteor CTD | $2\ 10^{-2}$ °C / Level 2 | $5\ 10^{-3}$ psu / Level 3 | $4\ \mu$mol/kg / Level 3 |
| R/V L'Atalante uCTD | $9\ 10^{-2}$ °C / Level 3 | $2\ 10^{-2}$ psu / Level 3 | |
| R/V Maria S. Merian uCTD | $9\ 10^{-2}$ °C / Level 3 | $2\ 10^{-2}$ psu / Level 3 | |
| R/V L'Atalante MVP | $3\ 10^{-2}$ °C / Level 3 | $1\ 10^{-2}$ psu / Level 3 | |
| R/V Maria S. Merian MVP | $4\ 10^{-2}$ °C / Level 3 | $2\ 10^{-2}$ psu / Level 3 | |
| UEA Gliders | $1.8\ 10^{-1}$ °C / Level 3 | $7\ 10^{-2}$ psu / Level 4 | |
| IFM Gliders | $1\ 10^{-1}$ °C / Level 3 | $4\ 10^{-2}$ psu / Level 3 | $28\ \mu$mol/kg / Level 3 |
| Kraken Glider | $3\ 10^{-2}$ °C / Level 3 | $1\ 10^{-2}$ psu / Level 3 | |
| Argo floats | $4\ 10^{-2}$ °C / Level 3 | $1.5\ 10^{-2}$ psu / Level 3 | $10\ \mu$mol/kg / Level 3 |
| Surface only | | | |
| | **Parameter uncertainty / Level of traceability** | | |
| **Platforms** | **Temperature** | **Salinity** | **Oxygen** |
| R/Vs TSG (without Meteor) | $2\ 10^{-2}$ °C / Level 2 | $4\ 10^{-2}$ psu / Level 2 | |
| R/V Meteor TSG | $2\ 10^{-2}$ °C / Level 3 | $4\ 10^{-2}$ psu / Level 3 | |
| Saildrones | $2.5\ 10^{-2}$ °C / Level 3 | $4\ 10^{-2}$ psu / Level 3 | |
| BSW Surface drifters | $2.5\ 10^{-2}$ °C / Level 3 | $2.5\ 10^{-2}$ psu / Level 3 | |
| BSC Surface drifters | $4\ 10^{-2}$ °C / Level 3 | $2\ 10^{-1}$ psu / Level 3 | |
| BRST Surface drifters | $2\ 10^{-2}$ °C / Level 3 | | |

**Table 1.** Table summarizing the uncertainties for each parameter measured by type of observation platform.

## 6 Data availability

The underlying primary CTD and TSG datasets (as well as the S-ADCP acquisitions), used in this study for comparison with other devices, are available for each R/V on the AERIS website: https://observations.ipsl.fr/aeris/eurec4a/#/ (Stevens et al., 2021). Additionally, for the R/V L'Atalante, the CTD measurements can be retrieved on the Seanoe website: https://doi.org/10.17882/79096 (Speich et al., 2021b, a). For the R/V Maria S. Merian, the CTD thermohaline measurements are referenced on the Pangea website: https://doi.pangaea.de/10.1594/PANGAEA.956057 (Karstensen and Krahmann, 2023b); the L-ADCP measurements, mounted on the CTD rosette from the R/V Maria S. Merian can be retrieved here: https://doi.pangaea.de/10.1594/PANGAEA.956063 (Karstensen and Krahmann, 2023a). Calibrated TSG measurements for the R/Vs Maria S. Merian and Meteor are also available on the Pangea website under the cruise names of MSM89: https://doi.pangaea.de/10.1594/PANGAEA.951515 (Karstensen et al., 2022) and M161: https://doi.pangaea.de/10.1594/PANGAEA.951515 (Mohr et al.,

2022) respectively (Karstensen et al., 2020; Mohr et al., 2020).

After the secondary data quality control process was applied, the various data sets were interpolated to the same vertical pressure grid with a 0.5 dbar resolution. This was done for temperature, salinity, and dissolved oxygen data and, when available, horizontal ocean currents. For each device, a NetCDF file is available for download on the Seanoe website, organized by type of
device and ship, with its own DOI. The naming of the variables and parameters follows the convention of the NetCDF Climate and Forecast (CF) Metadata Conventions in Eaton et al. (2003). The secondary quality control uCTD and MVP profiles can be found on the Seanoe website: https://doi.org/10.17882/91352 (L'Hégaret et al., 2020c) and https://doi.org/10.17882/91485 (L'Hégaret et al., 2020b) respectively. The original calibrated profiles from the R/V Maria S. Merian are also available on the Pangea website, for the uCTD: https://doi.pangaea.de/10.1594/PANGAEA.956139 (Karstensen and Krahmann, 2023d), and
for the MVP: https://doi.pangaea.de/10.1594/PANGAEA.956141 (Karstensen and Krahmann, 2023c).

Additionally, the concatenated sections described in the previous section, composed of CTD, secondary quality control uCTD and MVP profiles, and S-ADCP measurements from the R/Vs L'Atalante and Maria S. Merian can be accessed on the Seanoe website: https://doi.org/10.17882/92071 (L'Hégaret et al., 2020a).


Measurements from the autonomous devices, Saildrones, underwater gliders and drifters can also be retrieved from the AERIS website: https://observations.ipsl.fr/aeris/eurec4a/#/ (Stevens et al., 2021). Furthermore, the UEA gliders measurements are available on the British Oceanographic Data Center website: doi:10.5285/c596cdd7-c709-461a-e053-6c86abc0c127 (Rollo, 2021). The NASA Saildrones can be access or the dedicated website: https://doi.org/10.5067/SDRON-ATOM0 (Sail-
drone, 2020). Data from the Argo floats are available on the Coriolis website: https://dataselection.coriolis.eu.org/ (Argo, 2000). Table 2 summarizes the different links and DOIs to access these various datasets.

## 7   Conclusions

The oceanic instruments deployed during the EUREC[4]A-OA experiment provide a large set of observations characterized by
different oceanic structures, including mesoscale eddies intensified at the surface or at depth, finer scale filaments, and salinity barrier layers. Nevertheless, the wide variety of devices at our disposal require precise pre- and post-cruise calibrations and cross validations, so the data can be used to its full potential. In this study, we aimed at describing all sensors and their measurements, the sources of errors, and the methods used to correct them. Then, we ranked them, taking into account how they were validated and how they can be related to one another. The adopted strategy and the complementarity of the different ob-
servations enable descriptions and quantification of such processes with unprecedented detail. This underlines the importance of deploying CTD stations, and discussing their calibrations, our only way to compare water parcels sampled at depth with sensor measurements by performing close-by profiles with other devices or attaching probes to the rosette. With that, we are

| Vertical Profiles | | |
|---|---|---|
| **Platforms** | **Parameters** | **DOI/Access Data** |
| R/V L'Atalante CTD | Temperature/Salinity/Oxygen and Velocity | https://doi.org/10.17882/79096 |
| R/V Maria S. Merian CTD | Temperature/Salinity/Oxygen | https://doi.pangaea.de/10.1594/PANGAEA.956057 |
| R/V Maria S. Merian CTD | Velocity | https://doi.pangaea.de/10.1594/PANGAEA.956063 |
| R/V Meteor CTD | Temperature/Salinity | https://observations.ipsl.fr/aeris/eurec4a/#/ |
| R/Vs L'Atalante and Maria S. Merian uCTD | Temperature/Salinity | https://doi.org/10.17882/91352 |
| R/V Maria S. Merian uCTD | Temperature/Salinity | hthttps://doi.pangaea.de/10.1594/PANGAEA.956139 |
| R/Vs L'Atalante and Maria S. Merian MVP | Temperature/Salinity | https://doi.org/10.17882/91485 |
| R/V Maria S. Merian MVP | Temperature/Salinity | https://doi.pangaea.de/10.1594/PANGAEA.956141 |
| R/Vs L'Atalante and Maria S. Merian concatenated sections | Temperature/Salinity and Velocity | https://doi.org/10.17882/92071 |
| UEA Gliders | Temperature/Salinity | https://doi.org/10.5285/ c596cdd7-c709-461a-e053-6c86abc0c127 |
| IFM and Kraken Gliders | Temperature/Salinity/Oxygen | https://observations.ipsl.fr/aeris/eurec4a/#/ |
| Argo Floats | Temperature/Salinity/Oxygen | https://dataselection.coriolis.eu.org/ |
| Surface Only | | |
| **Platforms** | **Parameters** | **DOI/Access Data** |
| R/V L'Atalante TSG | Temperature/Salinity | https://observations.ipsl.fr/aeris/eurec4a/#/ |
| R/V Maria S. Merian TSG | Temperature/Salinity | https://doi.pangaea.de/10.1594/PANGAEA.951260 |
| R/V Meteor TSG | Temperature/Salinity | https://doi.pangaea.de/10.1594/PANGAEA.951515 |
| Saildrones | Temperature/Salinity and Velocity | https://doi.org/10.5067/SDRON-ATOM0 |
| Surface Drifters | Temperature/Salinity | https://observations.ipsl.fr/aeris/eurec4a/#/ |

**Table 2.** Table summarizing the DOIs and parameters measured by type of observation platform.

able to, at best, correct the measurements from other sensors, or at least, quantify their uncertainties.

We propose here a way of estimating the uncertainties by assessing the three main sources of variability between measurements on isopycnal levels or at the surface : background oceanic variability (lateral variability and internal wave field) and the sensor variability. The background oceanic variability is highly depth-dependent, thus for comparison we chose to compare profiles on isopycnal levels, only focusing on measurements performed below the mixing layer. Nevertheless, this method largely depends on the number of observations and their calibrations and validations using water samples. It also underlines
the importance of having synoptic profiles of the devices for comparison and hence, corrections. In the end, we make available a finalized dataset with calibrated and cross-validated thermohaline, chemical, and dynamical measurements, with their associated uncertainties after secondary QC.

*Author contributions.* Pierre L'Hégaret performed the secondadry quality control and adjustments to the uCTD, MVP and Gliders mea-
surements, their cross-calibration, and wrote this manuscript. Florian Schütte worked on the comparison of the CTD calibration methods from GEOMAR and IFREMER, and on the calibration of the uCTD profiles. Sabrina Speich, Gilles Reverdin, and Johannes Karstensen contributed to the mansucript writing and conceived of and led a major component of EUREC[4]A. Rémi Laxenaire, Gregory Foltz, Dongxiao Zhang made major contributions to the broader coordination and execution of scientific activities during and after the experiment. Corentin Subirade participated in the preparation and redaction of this manuscript. Karen J. Heywood, Elizabeth Siddle, and Callum Rollo deployed
the UEA gliders and processed their data. The R/V Meteor CTD data were obtained and processed by Darek Baranowski. The R/V Meteor ADCP data were undertaken onboard by Callum Rollo, and subsequently processed by Tim Fischer. Rena Czeschel processed the ADCP data from the R/Vs Maria S. Merian. Michael Schlundt processed the TSG data from the R/Vs Maria S. Merian and Meteor. Gerd Krahmann processed the CTD, uCTD, MVP, and glider (IFM03, 09, 12) data. Philippe Le Bot and Stéphane Leizour processed the CTD and uCTD data on board of the R/V L'Atalante. Caroline Le Bihan processed the CTD data from the R/V L'Atalante and performed its calibration.

*Competing interests.* No competing interests are present.

*Acknowledgements.* This research has been supported by the people and government of Barbados; by the European Research Council (ERC) advanced grant EUREC[4]A (grant agreement no. 694768) under the European Union's Horizon 2020 research and innovation program (H2020), with additional support from CNES (the French National Center for Space Studies) through the TOSCA SMOS-Ocean, TOEddies, and EUREC[4]A-OA proposals, the French national program LEFE INSU, IFREMER, the French research fleet, the French re-
search infrastructures AERIS and ODATIS, IPSL, the EUREC[4]A-OA JPI Ocean and Climate program, the Chaire Chanel program of the Geosciences Department at ENS, Météo-France, by the Max Planck Society and its supporting members and by the German Research Foundation (DFG) and the German Federal Ministry of Education and Research (grant nos. GPF18-1-69 and GPF18-2-50). DBB was supported by the Poland's National Science Centre (grant no.UMO-2018/30/M/ST10/00674). We also acknowledge the mesoscale calculation server

CICLAD http://ciclad-web.ipsl.jussieu.fr dedicated to Institut Pierre Simon Laplace modeling effort for technical and computational support.

We also warmly thank the captain and crew of RVs Atalante, Maria S. Merian, Meteor and Ronald H. Brown. Gregory Foltz was supported by the CVP Program of NOAA's Climate Program Office, and by base funds to NOAA/AOML.

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

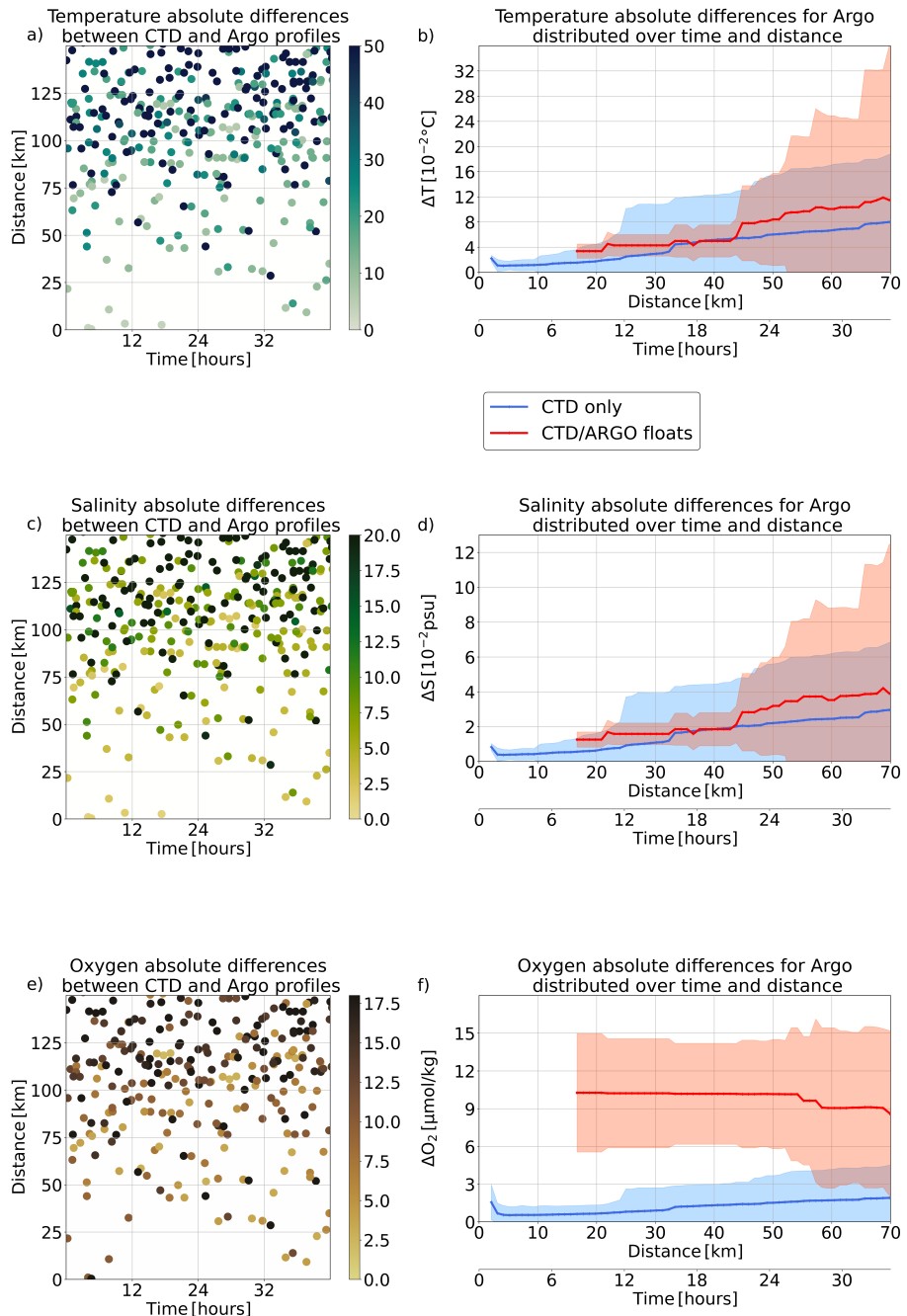

**Figure 19.** a) Absolute temperature differences between CTD/Argo floats pairs of profiles on isopycnal levels and averaged vertically distributed in time and distance. Each pair is composed of one CTD and one Argo float profiles. b) Temperature differences as a function of time and distance for pairs of profiles. In blue for pairs of CTD only profiles, in red for pairs composed of one CTD and one Argo float profiles. The middle line of panels is the same as above, but for absolute salinity differences. The bottom line of panels correspond to absolute oxygen differences.

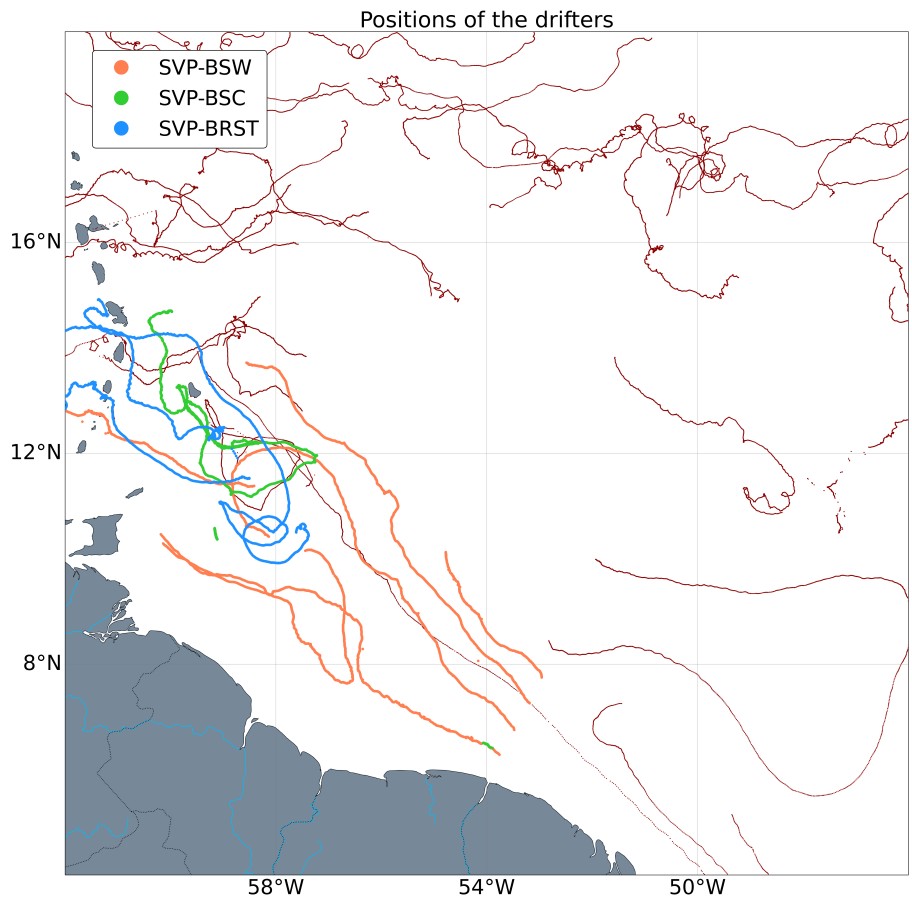

**Figure 20.** Map of the drifters positions from December 2019 to May 2020. Thick colored lines display the drifters that have been deployed from the R/V L'Atalante, while thin red lines shows the other drifters observed in the region during the same period.

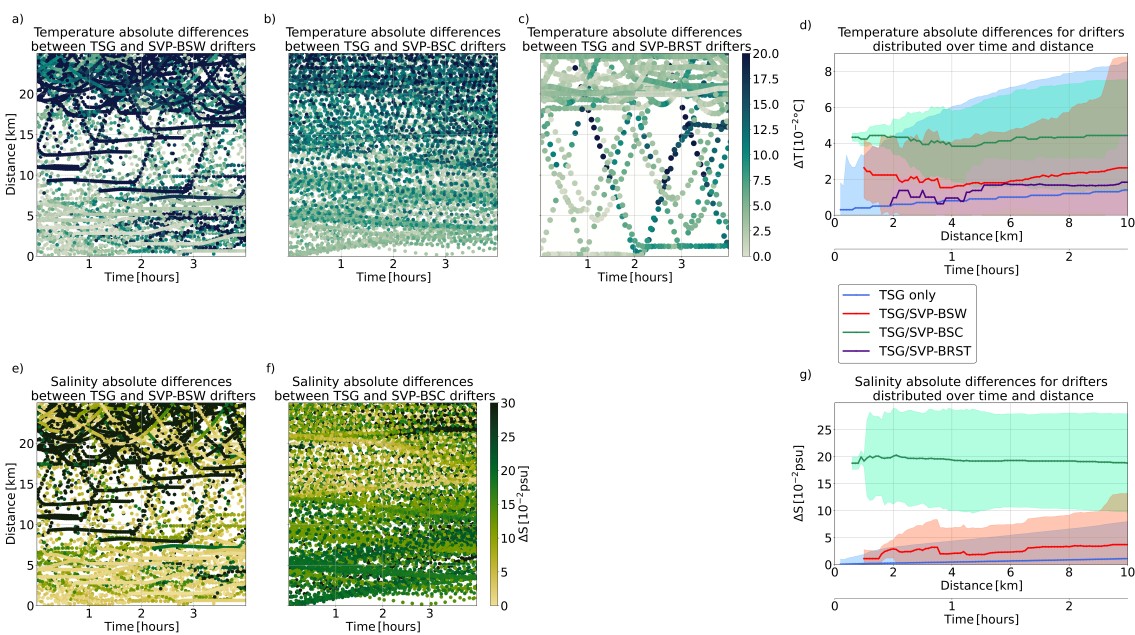

**Figure 21.** a), b), and c) Absolute temperature differences between TSG/drifters pairs of measurements distributed in time and distance. Each pair is composed of one R/V's TSG measurement and a drifter's one. Respectively a), b), and c), correspond to the SVP-BSW, SVP-BSC, and SVP-BRST drifters. d) Temperature differences as a function of time and distance for a pair of surface measurements. In blue for pairs of CTD only profiles, in red for pairs composed of one TSG and one SVP-BSW measurements, in green for pairs composed of one TSG and one SVP-BSC measurements, and in purple for pairs composed of one TSG and one SVP-BRST measurements. e) to g), same as the above line, but for absolute salinity differences without the SVP-BRST drifters.

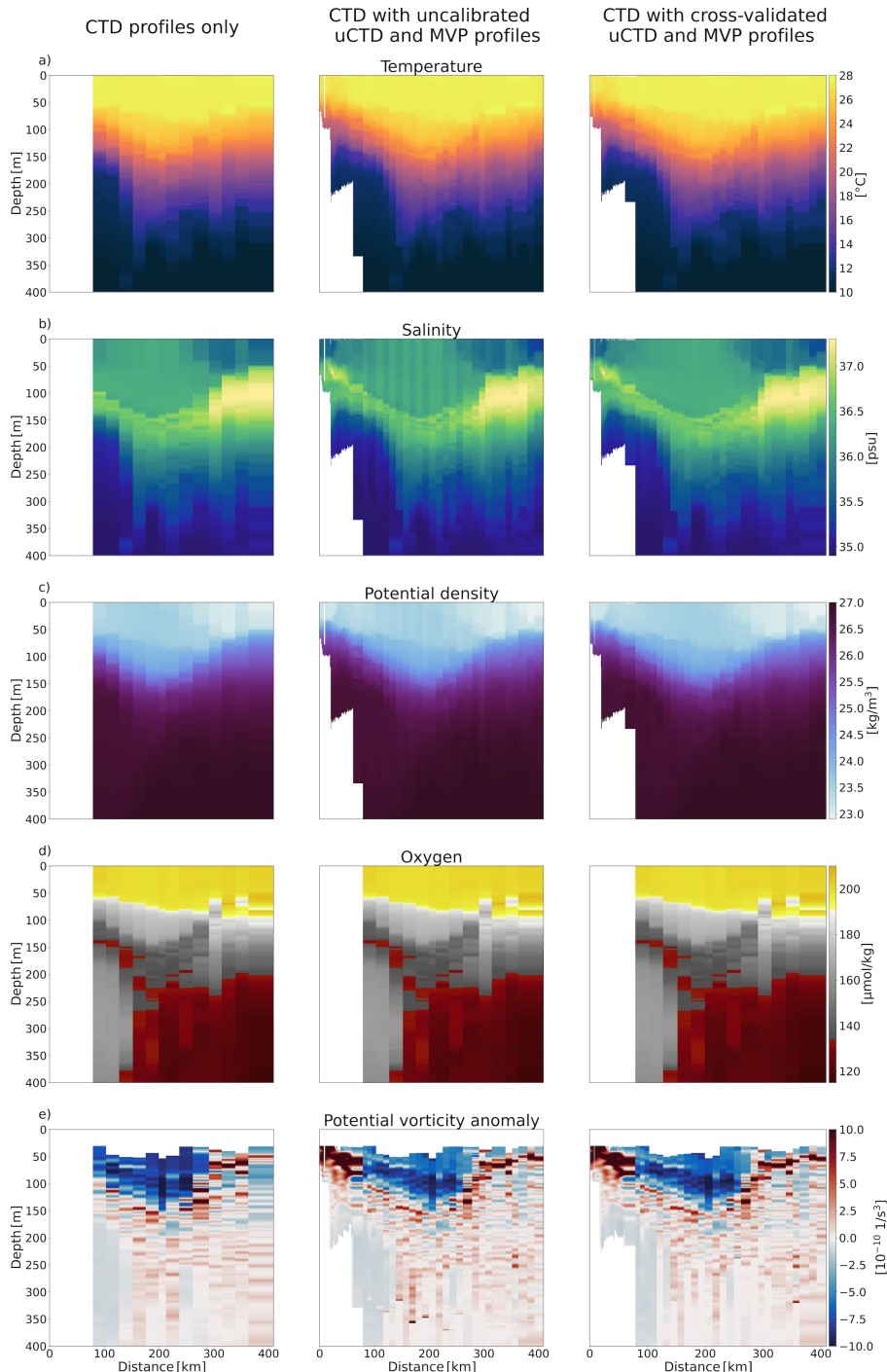

**Figure 22.** Vertical sections of thermohaline and dynamic characteristics performed by the R/V L'Atalante. The first column shows the section with only the CTD profiles, on the second column the uncalibrated profiles of MVP and uCTD are added, the third column is the same but for calibrated profiles. a) is for temperature, b) for salinity, c) for potential density, d) for oxygen (only CTD profiles measured oxygen), and e) for Ertel Potential Vorticity based on the ADCP measurements (not shown).