# Peer review of "Ocean cross-validated observations from the R/Vs L'Atalante, Maria S. Merian and Meteor and related platforms as part of the EUREC4A-OA/ATOMIC campaign"

_Earth System Science Data, 2022_

## Author Comment (AC1)

**General comments**

The manuscript describes an impressive observational effort in the tropical western North Atlantic. Four research vessels joint in the EUREC4A-OA experiment and conducted measurements in the region, when also deploying autonomous vehicles. A wealth of data was acquired, based on a multitude of different sensors.

I really appreciate the effort taken in this manuscript; combining data of various sources in order to assess their quality needs thorough procedures and considerations, as described here. Nevertheless, I think the manuscript should be improved in a few points:

- Make sure to address clearly which data are considered here and which are not. Especially, say at the beginning of the manuscript that no data from US Ronald H. Brown were analyzed and less data from Meteor were available (e.g. no L-ADCP, uCTD). Meteorological sensors are not addressed. It is valuable to inform the reader that EUREC4A-OA included more measurements than presented here, but it is also necessary to give an overview of the data analyzed here at the very beginning. Readers need to know what to expect.
- Additionally, the point that no lab samples for salinity and oxygen were available for CTD calibration on Meteor must be placed more prominently. Maybe it is possible to say why the salinity samples were not analyzed? It is very unfortunate. However, especially in the Calibration and Observation chapters (Sections 2 and 3) text is written as if procedures apply to all three vessels. This is not the case. See more specific comments below.
- Following the point above, it would be interesting to see a comparison of the profiles which were taken nearby the Meteor CTD (In 194-196). Figure 4 is not sufficient in addressing the error of the Meteor CTD. The description needs a figure with all profiles conducted in close proximity, with a focus on the deeper part of the water column where variability is low.
- The figures generally include the distance class up to 100 km. I would not expect much similarity in profiles with so much distance between them. I think the more informative class of up to 25 km distance should get more space and the ranges of the axes in the figures should be zoomed to a smaller range. In other words, I don't get what we learn from the large distance comparison, no conclusion regarding sensor error is possible from these pairs. Maybe we need them for Argo, where distances and errors are larger in general, but for CTD, where errors are small and reliability is high, I recommend to focus on the profiles which are close to each other, even when there are only few pairs.
- Figures of the type Fig. 14 d,h,j should include information on the amount of data used to calculate the mean and standard deviation.
- References are often too short, no doi is given and sometimes there is no hint on where to find the papers (e.g. Otosaka et al. or Branellec et al.). Some should be available online, please provide links. Is Le Bot et al. only available in French? Please provide an alternative if possible.

Since the lists above and below are rather long, I suggest a major revision of the manuscript.

**Answer to General comments**

We would like to thank the reviewer for the in-depth reading of this draft, and for the comments that will help improve the manuscript. Hereafter, we will address the major points and specify how we changed the text accordingly.

**Comment:** « Make sure to address clearly which data are considered here and which are not. Especially, say at the beginning of the manuscript that no data from US Ronald H. Brown were

analyzed and less data from Meteor were available (e.g. no L-ADCP, uCTD). Meteorological sensors are not addressed. It is valuable to inform the reader that EUREC4A-OA included more measurements than presented here, but it is also necessary to give an overview of the data analyzed here at the very beginning. Readers need to know what to expect. »

**Answer:** We have added a paragraph in the introduction, after the experiment presentation and before the secondary quality control description, to specify on which platforms we are focusing and on which measurements. Namely on the devices measuring the ocean temperature, salinity, oxygen, and velocity at the surface and at depth.

**Comments:** « Additionally, the point that no lab samples for salinity and oxygen were available for CTD calibration on Meteor must be placed more prominently. Maybe it is possible to say why the salinity samples were not analyzed? It is very unfortunate. However, especially in the Calibration and Observation chapters (Sections 2 and 3) text is written as if procedures apply to all three vessels. This is not the case. See more specific comments below. » and « Following the point above, it would be interesting to see a comparison of the profiles which were taken nearby the Meteor CTD (In 194-196). Figure 4 is not sufficient in addressing the error of the Meteor CTD. The description needs a figure with all profiles conducted in close proximity, with a focus on the deeper part of the water column where variability is low. »

**Answer:** On the R/V Meteor no salinometer were available, so no calibration was performed onsite. Samples were taken and transferred to GEOMAR after the cruise, but too few samples were available to perform the analysis. The calibration of temperature and salinity was based on two factors, the cross calibration between the primary and secondary sensors, and cross validation done with close-by stations from the R/V Maria S. Merian. Particularly one cast was performed side by side. Oxygen was not tested. We have integrated this information to the draft in section 3.1.1 and 3.1.2 focusing on the description of the CTD measurements.

**Comment:** « The figures generally include the distance class up to 100 km. I would not expect much similarity in profiles with so much distance between them. I think the more informative class of up to 25 km distance should get more space and the ranges of the axes in the figures should be zoomed to a smaller range. In other words, I don't get what we learn from the large distance comparison, no conclusion regarding sensor error is possible from these pairs. Maybe we need them for Argo, where distances and errors are larger in general, but for CTD, where errors are small and reliability is high, I recommend to focus on the profiles which are close to each other, even when there are only few pairs. »

**Answer:** Indeed, at first the choice of scale was chosen to represent a large number of pairs of profiles, but the interest is higher with closer measurements. We have modified the figures accordingly. We are still showing up to 40/50 km in order to also compare the devices with the « CTD only pairs », because below 25 km too few of them are found to be statistically significant.

**Comment:** « Figures of the type Fig. 14 d,h,j should include information on the amount of data used to calculate the mean and standard deviation. »

**Answer:** We have added an information on the number of measurements used in the different paragraphs while commenting these Figures.

**Comment:** « References are often too short, no doi is given and sometimes there is no hint on where to find the papers (e.g. Otosaka et al. or Branellec et al.). Some should be available online, please provide links. Is Le Bot et al. only available in French? Please provide an alternative if possible. »

**Answer:** We have revised the references, the DOI and links to access them. Sadly Le Bot et al. is only available in French.

**Specific comments**

We have implemented the typos and suggestions made by the reviewer. Hereafter we will answer to the questions.

**Comment:** «Ln 45: Assuming that the lab at IFREMER is shallower than 2000 m, the derived, uncertainty is probably for a certain pressure, thus 2000 dbar? »

Answer: Indeed, it was for a certain pressure and not depth, it has been modified.

Comment: « Ln 151: no lab calibration was done before and after (?) the cruise »

Answer: Indeed, we have specified that calibration was instead performed at regular interval.

**Comment:** « *Ln* 248-250: move to the beginning or very end of the subsection, it seems out of place here. Why -9? »

Answer: The value -9 was used to easily distinguish them from the other devices

**Comment:** « *Ln* 279: *It is an important result that the methods agree in the final dataset. Please give more details on "no major differences" and add a figure, showing calibrated profiles from the two toolboxes.* »

**Answer:** We have added a figure focusing on the comparison between the toolboxes and a paragraph discussing their differences. Overall we observe that the differences, on average, remain in the same order of magnitude as the sensors' accuracy, and that they are maximized around the thermocline.

**Comment:** « *Ln* 294: *it would be nice to have a table listing the final uncertainties. What uncertainty was assigned to the Meteor data?* »

Answer: The uncertainties have been added to a table summarizing all the devices.

**Comment:** « *Ln* 476 *ff: I* cannot follow the conclusion here. When the profiles of the pairs are in close vicinity to each other, variability should be small. Cannot be the missing calibration of the Meteor CTD (no salinity samples) be the cause of the large differences for the UEA gliders, which are mainly based on comparisons with Meteor? »

**Answer:** Indeed, we have modified the paragraph accordingly, stating that the comparisons are mostly made with Meteor CTD.

Comment: « Ln 490: was the bias corrected? What is the final uncertainty and level? »

**Answer:** The bias is not directly corrected since we cannot attribute its origin to the CTD uncertainty or to the underwater gliders. The final uncertainty and level are added to the text and synthesize in a table.

**Anonymous Referee #2**

**General comments**

This study shows the data acquired during the EURECA4-OA/ATOMIC experiments in January and February 2020 in the northwestern Tropical Atlantic. The experiment was carried out using four oceanographic vessels and various autonomous platforms resulting in an extensive dataset, which is definitely an important contribution to the oceanographic community. The work is good and the authors made a big effort in this manuscript but in some parts of the text the information is confusing or needs to be more accurate. My major concerns are:

- In the introduction section I miss the goal to acquire this data, why is it relevant? I suggest adding a little paragraph about this region in oceanographic terms (i.e. the ocean circulation system we are observing).

- In the introduction, the authors mention the R/V Ronald H. Brown as it would be part of the data described in the rest of the manuscript but it's not, it looks like only data acquired from German and French vessels are used. I think a better distinction needs to be made between the context of the experiment and the data used in this manuscript.

- There is a lot of information and sometimes the text is dense. I suggest providing the reader with more schematic information in some parts of the text. For instance, the authors could add one or two more tables with the type of instruments used at each vessel, sensors, errors, number of profiles, the level of hierarchy attributed, etc...

- The authors state in section 3.2 that the largest salinity differences between CTD pairs are found near the surface, but I do not agree with what is shown in Figure 4, wherein generally it is observed that these differences are larger in the thermocline layer compared to the mixing layer. Because of this, I think it is not very accurate to remove the first 50 meters to make the calculations in figure 5. One possibility could be to make the calculations in figure 5 as well as the next ones by depth or density ranges. Furthermore, I do not understand the need to show the differences beyond 60 or 80 km as it seems to me that the distances are too large to make a comparison between CTDs.

**Answer to General comments**

We thank the reviewer for the detailed reading of this manuscript and underlining its weaknesses. We have answered all the general and specific comments and modified this draft accordingly, and we think it improved overall the comprehension of our study. Please find hereafter our detailed answers.

**Comments:** « In the introduction section I miss the goal to acquire this data, why is it relevant? I suggest adding a little paragraph about this region in oceanographic terms (i.e. the ocean circulation system we are observing). » And « In the introduction, the authors mention the R/V Ronald H. Brown as it would be part of the data described in the rest of the manuscript but it's not, it looks like only data acquired from German and French vessels are used. I think a better

distinction needs to be made between the context of the experiment and the data used in this manuscript. »

**Answer:** We have modified the introduction, adding a paragraph on the motivations and dynamics of the region of interest. We have also described in more details on which datasets this study focuses on.

**Comment:** « There is a lot of information and sometimes the text is dense. I suggest providing the reader with more schematic information in some parts of the text. For instance, the authors could add one or two more tables with the type of instruments used at each vessel, sensors, errors, number of profiles, the level of hierarchy attributed, etc... »

**Answer:** We have added a table summarizing the different devices for each R/V as well as their level of traceability and uncertainty. At the end of the introduction we have also specified that for each devices we first focus on the number of measurements, the characteristics of the sensors, how they are calibrated, their level in the hierarchy of traceability, and the associated uncertainty.

**Comment:** « The authors state in section 3.2 that the largest salinity differences between CTD pairs are found near the surface, but I do not agree with what is shown in Figure 4, wherein generally it is observed that these differences are larger in the thermocline layer compared to the mixing layer. Because of this, I think it is not very accurate to remove the first 50 meters to make the calculations in figure 5. One possibility could be to make the calculations in figure 5 as well as the next ones by depth or density ranges. Furthermore, I do not understand the need to show the differences beyond 60 or 80 km as it seems to me that the distances are too large to make a comparison between CTDs. »

**Answer:** Indeed, as stated by the reviewer, the largest differences are found at depth. We have changed these Figures to include all the water column, with the differences calculated on same density levels and then averaged vertically. Additionally, we have modified the Figures to show a distance range of 0 to 40/50 km for comparison of instruments with the CTD profiles. Further than that the variability increases too much, as shown on Figure 4. We have also added, in the same section, a paragraph discussing the different processing from IFREMER and GEOMAR, and how they impact the final parameters.

**Specific comments**

We have implemented the typos and suggestions made by the reviewer. Hereafter we will answer to the questions.

Comment: « L27. Can you show the two subregions in Figure 1? »

Answer: We have modified this figure.

**Comment:** « L229. I suggest to avoid the nomenclature Master-Slave (https://www.nytimes.com/ 2021/04/13/technology/racist-computer-engineering-terms-ietf.html) »

**Answer:** Thank you for underlining this problem, indeed this kind of nomenclature must be avoided.

Comment: « L498-520. Please, specify from which vessel the saildrones are deployed. »

**Answer:** We have specified that the Saildrones were deployed from Barbados.

**Comment:** « L543-546. Do the authors need a separate section for dissolved oxygen? It is only 3 lines section. »

**Answer:** We have merged this subsection with the previous one.

Comment: « L556. looping what? eddies? inertial oscillations? »

**Answer:** We have specified that they loop inside mesoscale eddies.

---

## Author Response (AR2)

**Comment from the Editor**

*I do not consider all of referee #1's "specific comments" to be answered. This applies to Ln28-33; Ln161; Ln216ff; Ln266; Ln277; Ln314; Ln407; Ln458; Ln460. Please respond to these comments, which are not simple corrections but comments on the content of the text passages.*

**Answer to Editor**

Indeed our answers to these specific comments from the first reviewer were quite short. We did modified our manuscript accordingly, but these information were not stated clearly. Hereafter we will expand on our answers and how we modified the manuscript.

**Comment:** *« Ln 28-33: the paragraph seems a little out of place and focus. It is neither a full description of the working region, nor a broad overview. Add a sentence on the importance of the AMOC for the interhemispheric exchange, connecting the tropical South and North Atlantic, and the role of NBC rings. Give references (the only reference to a paper on the region is given in line 290, which is out of place as well). A map with currents would be helpful as well. The division in two regions "east of Barbados" and "to the south" is not conclusive either, what about the rest of the box in Figure 1 (north and west of Barbados)? »*

**Answer:** This paragraph has been removed to be replaced by a more complete description of the region. It can be found from line 29 to 45 of the corrected version of the manuscript. A statement has been added on the inter hemispheric exchange, with also a citation from Johns (Elsevier Oceanography Series, 2003). We have also modified Figure 1 to add a schematic representation of the main surface currents of our two subregions of interest, represented with two boxes instead of one. Moreover, we have specified which R/Vs cover these subregions.

**Comment:** *« Ln 161: I guess there were also two SBE4 sensors? Please clarify. »*

**Answer:** Indeed it was two SBE4 sensors that were mounted on the CTD rosette. This has been add in the corrected manuscript (line 192).

**Comment:** *« Ln 216 ff: What about Meteor? Did the Meteor CTD measure oxygen, with one or two SBE43 sensors? What happened to the data? »*

**Answer:** We have added information on which R/Vs measured oxygen. These modifications can be found lines 242 and 252. Only the R/Vs L'Atalante and Maria S. Merian measured oxygen, no measurements nor samples were collected for the R/V Meteor.

**Comment:** *« Ln 266: does this statement refer to all measurements described in this subsection or to PAR sensors only? »*

**Answer***:* This statement applies to all measurements, it has been corrected line 295.

**Comment:** *« Ln 277: I do not understand why the positioning of the CTD station was different. Please explain. »*

**Answer:** The posting of each CTD station is defined differently. From IFREMER it is defined as the location and time of first measurement, while for GEOMAR it is defined as on average over the period when the CTD is deployed. This statement can be found from line 306 to 308 of the corrected manuscript.

**Comment:** *« Ln 314 ff: what about Meteor? Were water samples taken and analyzed? »*

**Answer:** It only concerns the R/VS L'Atalante and Maria S. Merian. Thanks to the reviewer's comments, in the corrected manuscript, we have clearly specified which R/Vs are concerned,

mainly in Section 3.1.1, but also in all the text. The R/V Meteor was calibrated by comparing its measurements with close-by dedicated CTD profiles performed by the R/V Maria S. Merian (Lines 349-351).

**Comment:** *« Ln 407: what was the cut-off frequency? »*

**Answer:** The cut-off frequency is 0.2Hz (Line 457).

**Comment:** *« Ln 458: "all were equipped with Seabird CTDs" contradicts line 448 (Kraken). What are the uncertainties for the IFM and UEA gliders? »*

**Answer:** Indeed, we corrected the text. While the Kraken was equipped with a GPCTD (pumped), the IFM and UEA ones were equipped with SBE41 (unpumped) CTD (Line 510). IFM gliders have uncertainties of 0.1°C, and $4 \cdot 10^{-2}$ psu, and the UEA ones of $1.8 \cdot 10^{-1}$°C and $7 \cdot 10^{-2}$ psu. These information can be found in the strongly modified paragraph between lines 522 and 536, as well as in Table 1.

**Comment:** « Ln 460: "some devices" – please specify which devices. »

**Answer:** This sentence was removed from the manuscript to specify that we were mentioning the Kraken glider.